# U-MARVEL: Unveiling Key Factors for Universal Multimodal Retrieval via Embedding Learning with MLLMs

**Xiaojie Li**[1,2*]    **Chu Li**[3*]    **Shi-Zhe Chen**[1†‡]    **Xi Chen**[1‡]
[1]Tencent PCG    [2]Nanjing University    [3]ByteDance
{chaxjli,shizhechen,jasonxchen}@tencent.com, lichu@bytedance.com

## Abstract

Universal multimodal retrieval (UMR), which aims to address complex retrieval tasks where both queries and candidates span diverse modalities, has been significantly advanced by the emergence of MLLMs. While state-of-the-art MLLM-based methods in the literature predominantly adopt contrastive learning principles, they often differ in their specific training recipes. Despite their success, the mechanisms underlying their retrieval capabilities remain largely unexplored, potentially resulting in suboptimal performance and limited generalization ability. To address these issues, we present a comprehensive study aimed at uncovering the key factors that drive effective embedding learning for UMR using MLLMs. We begin by implementing a general MLLM-based embedding learning pipeline, and systematically analyze the primary contributors to high-performing universal retrieval systems. Based on this, we explore various aspects of the details in embedding generation and training strategies, including progressive transition, hard negative mining and re-ranker distillation. Notably, our findings reveal that often-overlooked factors can have a substantial impact on model performance. Building on these discoveries, we introduce a unified framework termed U-MARVEL (**U**niversal **M**ultimod**A**l **R**etrie**V**al via **E**mbedding **L**earning), which outperforms state-of-the-art competitors on the M-BEIR benchmark by a large margin in supervised settings, and also exhibits strong zero-shot performance on several tasks such as composed image retrieval and text-to-video retrieval. These results underscore the generalization potential of our framework across various embedding-based retrieval tasks. Code is available at https://github.com/chaxjli/U-MARVEL.

## 1 Introduction

Multimodal information retrieval (Liu et al., 2022) has emerged as a pivotal research direction at the intersection of multimodal understanding and information retrieval. It serves as a cornerstone for a wide range of modern AI applications, including retrieval-augmented generation (RAG) (Jiang et al., 2023; Cong et al., 2023), text-image retrieval (Baldrati et al., 2023; Zhang et al., 2024b), and visual question answering (VQA) (Chow et al., 2024; Chun et al., 2021). While existing methods such as CLIP (Radford et al., 2021), BLIP (Li et al., 2022; 2023a), and CoCa (Yu et al., 2022) have demonstrated impressive performance in cross-modal retrieval, they often struggle to address the diverse and complex requirements encountered in real-world scenarios, ranging from fine-grained instruction following to multi-turn interleaved interactions.

To tackle these challenges, the research community has increasingly focused on universal multimodal retrieval (UMR) (Wei et al., 2024), depicted in Fig. 1(a), which aims to develop unified, instruction-guided retrievers capable of handling diverse retrieval tasks across multiple modalities. Recent advancements in this area, such as LamRA (Liu et al., 2024b), MM-Embed (Lin et al., 2024),

---

[*]Equal contribution, work done at Tencent PCG.

[†]Project lead.

[‡]Corresponding authors.

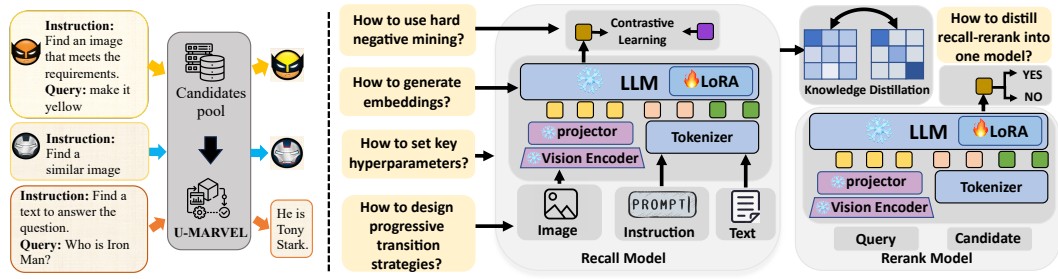

(a) Universal Multimodal Retrieval      (b) The Framework for U-MARVEL

Figure 1: U-MARVEL Exploration: We conduct a comprehensive study on effective design principles for embedding model architectures and optimal strategies for training embedding models.

GME (Zhang et al., 2024d), UniME (Gu et al., 2025) and VLM2VEC (Jiang et al., 2024b), have shown promising results on UMR powered by MLLMs. However, most of the existing approaches often directly adapt MLLMs to embedding tasks without systematically examining training schemes tailored for MLLM-based embedding models. Consequently, the impact of design decisions on retrieval performance remains inadequately understood, highlighting the need for a more systematic investigation—a gap that motivates our study.

In this paper, we aim to address these issues by systematically exploring the design principles for building high-performing universal multimodal retrievers with pre-trained MLLMs (Fig. 1(b)). To this end, we first implement a simple yet effective embedding learning pipeline based on contrastive learning, following the existing approaches in the literature. Subsequently, we conduct a comprehensive study to investigate three key aspects for training high-performance retrieval models built on MLLMs. Specifically, we explore: (1) how to adapt decoder-only MLLMs into instruction-guided embedding models; (2) how to train MLLM-based embedders within the context of a contrastive learning framework; and (3) how to distill the recall-then-rerank paradigm into a single model to further enhance retrieval performance. Our findings highlight several overlooked factors critical to retrieval performance, including the superiority of bidirectional attention with mean pooling over last-token mechanisms, and the interactions among parameters of batch size, learning rate, and temperature coefficient in contrastive learning. We also discover that integrating hard negative mining and knowledge distillation effectively mitigates computational inefficiencies in the recall-then-rerank pipeline, while simultaneously improving retrieval efficiency and accuracy.

Motivated by these insights, we present U-MARVEL, a unified framework for learning universal multimodal embeddings tailored to UMR tasks. Our method achieves significant improvements over state-of-the-art UMR approaches on the M-BEIR benchmark (Wei et al., 2024) in supervised settings, and also demonstrates strong zero-shot generalization abilities on tasks such as text-to-video retrieval and composed image retrieval.

The contributions of this paper can be summarized as follows. (1) We conduct a comprehensive exploration of the design space for MLLM-based universal retrieval, uncovering key factors that improve performance and providing actionable insights for future research. The findings presented in this paper aim to advance the effectiveness of UMR tasks within the research community. (2) We introduce U-MARVEL, a unified framework that achieves state-of-the-art performance in both supervised and zero-shot settings, showcasing its effectiveness across a wide range of retrieval tasks.

## 2 RELATED WORK

**Multimodal Large Language Models.** Large language models (LLMs) (Achiam et al., 2023; Yang et al., 2024; Liu et al., 2024a; Touvron et al., 2023), including multimodal LLMs (Zhang et al., 2024c; Yao et al., 2024), have achieved remarkable success across a wide range of general-purpose tasks. For instance, LLaVA (Liu et al., 2023a) integrates image and text processing for tasks like visual question answering, image description, and visual reasoning. The Qwen-VL series (Wang et al., 2024a; Bai et al., 2025) enhances this by supporting flexible image resolutions and extending

capabilities to video understanding through temporal modeling and localization. InternVL3 (Zhu et al., 2025) unifies multimodal and linguistic learning by leveraging both multimodal data and text corpora in a single pre-training stage.

**Multimodal Retrieval.** Multimodal retrieval aims to align diverse modalities (e.g., text, images, or their combinations) within a shared embedding space for semantic similarity matching. Early works (Radford et al., 2021; Li et al., 2022; Yu et al., 2022) leveraged large-scale contrastive learning to align image-text pairs, enabling zero-shot classification and cross-modal retrieval, but struggled with complex multimodal inputs. Recent advancements have addressed these limitations through innovative approaches based on MLLMs (Jiang et al., 2024a; Wei et al., 2024; Jiang et al., 2024b; Lan et al., 2025). LamRA (Liu et al., 2024b) adapts MLLMs into retrievers using lightweight LoRA modules with joint pointwise and listwise reranking; MM-Embed (Lin et al., 2024) improves alignment via MLLM fine-tuning and modality-aware hard negative mining; GME (Zhang et al., 2024d) develops a unified multimodal retrieval framework leveraging MLLMs with synthetic data augmentation; and UniME (Gu et al., 2025) combines textual knowledge distillation from LLM teachers with hard negative-enhanced instruction tuning, achieving robust cross-modal alignment for diverse retrieval tasks. While these works provide a strong foundation for UMR, the design space for MLLM-based embedding models remains underexplored. Unlike LLMs that generate outputs via autoregressive decoding, embedding models encode features of the entire input sequence, necessitating specialized strategies for feature extraction and tailored training methodologies. This work addresses these gaps through a systematic investigation of the design decisions for MLLM-based embedding models.

## 3 RECIPE FOR BUILDING U-MARVEL

In this section, we conduct a comprehensive study to identify the key factors that drive effective embedding learning for UMR through MLLMs. Following the mainstream approaches, we first establish a simple pipeline. Unless stated otherwise, we finetune the pre-trained Qwen2-VL-7B-Instruct (Wang et al., 2024a) as the embedding model with LoRA (Hu et al., 2022) by contrastive learning on M-BEIR dataset (Wei et al., 2024). Specifically, the InfoNCE loss (Oord et al., 2018) is selected as our training objective:

$$\mathcal{L}_{\text{InfoNCE}} = -\log \frac{\exp(\text{sim}(e_q, e_c^+)/\tau)}{\sum_i \exp(\text{sim}(e_q, e_{c^i})/\tau)} \tag{1}$$

where $e_q$, $e_c^+$, $e_{c^i}$ represent the embeddings of the query, the positive candidate, and any sample from the candidate set, respectively. $\tau$ is the temperature parameter. $\text{sim}(\cdot, \cdot)$ means the cosine similarity; More details are given in Appendix C. Throughout the ablation studies, we report the Average Recall as the evaluation metric, which is computed as the mean of Recall@5 or Recall@10 across the benchmark tasks (c.f. Appendix A). We then explore three major axes of design decisions in the following subsections:

- We investigate the adaptation of decoder-only MLLMs into instruction-aware embedders.
- We explore how to train embedding models within the framework of contrastive learning.
- We validate that the retrieval performance can be further enhanced through distillation from the recall-then-rerank paradigm.

### 3.1 HOW TO ADAPT MLLMS INTO EMBEDDING MODELS?

LLMs generate output tokens in an autoregressive manner, whereas embedding models encode holistic representations of input sequences, reflecting fundamental differences in their training objectives. Adapting LLMs into embedding models thus requires specialized training strategies. In this subsection, we examine this adaption from three aspects as follows.

#### 3.1.1 EMBEDDING EXTRACTION

> **Finding 1**: Generating embeddings with bidirectional attention and mean pooling outperforms the common approach of using compression prompts with the last token mechanism.

We first investigate the method for extracting embeddings of the token sequence of multimodal queries using MLLMs. Following prior approaches, the instruction is directly concatenated with the query. Existing methods can basically be divided into two categories based on the way embeddings are extracted. (1) **Last token**: previous works (Liu et al., 2024b; Jiang et al., 2024b; Zhang et al., 2024d; Jiang et al., 2024a; Gu et al., 2025) typically employ a prompt for compression. Generally, for multimodal input, the prompt is "*<image><text>...<image><text>Summarize above image and sentence in one word: emb*", where *<image>* and *<text>* are the placeholders. In this way, the feature of last token (i.e., *<emb>*) is used as the output embedding of the input sequence. (2) **Mean token**: an alternative method, adopted by MM-Embed (Lin et al., 2024), transforms the unidirectional attention mechanism into a bidirectional one and applies mean pooling to the features from the last hidden states to derive the embedding for the entire sequence.

Table 1: Comparison of last token and mean token mechanisms

| ID | Causal or Bidirectional | Last token or Mean token | Compression Prompt ("in one word") | Local Avg. | Global Avg. |
|----|-------------------------|--------------------------|------------------------------------|------------|-------------|
| 0  | Causal                  | Last token               | ✓                                  | 56.6       | 54.8        |
| 1  | Bidirectional           | Last token               | ✓                                  | 55.5       | 52.6        |
| 2  | Causal                  | Mean token               | ✓                                  | 33.7       | 27.6        |
| 3  | Bidirectional           | Mean token               | ✓                                  | 51.7       | 46.1        |
| 4  | Bidirectional           | Mean token               | ✗                                  | 57.2       | 55.2        |

We conducted detailed comparative experiments, as presented in Table 1. (1) The experimental results reveal that the effectiveness of the compression prompt exhibits dual dependency. On one hand, it establishes a strong coupling with the last token mechanism, as evidenced by ID-0 and ID-2, where replacing last token with mean token results in a significant performance drop of 22.9%/27.2%. On the other hand, causal attention enhances the impact of the compression prompt (ID-0 vs. ID-1) demonstrating a 1.1%/2.2% improvement over bidirectional attention. (2) Interestingly, bidirectional attention partially decouples the dependency between the compression prompt and the last token mechanism. For instance, ID-3 shows an 18%/18.5% improvement compared to ID-2. However, when combined with mean pooling, bidirectional attention remains constrained by the limitations of the compression prompt. (3) Furthermore, when compression prompt is removed from bidirectional attention with mean token mechanism (ID-4), it achieves a modest improvement of 0.6%/0.4% over ID-0, which is commonly used in the literature.

In conclusion, generating embeddings using bidirectional attention combined with mean pooling from the sequence demonstrates superior performance compared to the commonly employed approach of combining compression prompts with the last token mechanism. This difference may stem from the last token embedding being affected by recency bias, leading to an excessive reliance on the output of the final token. Notably, this finding aligns with the conclusions of NV-Embed (Lee et al., 2024), yet diverges from those presented in GME (Zhang et al., 2024d), offering a distinct perspective on architectural design and performance optimization.

### 3.1.2 INSTRUCTION INTEGRATION

**Finding 2**: Masking instruction tokens during mean pooling enhances embedding performance.

Motivated by prior studies on text embeddings (Lee et al., 2024; BehnamGhader et al., 2024), we mask out the instruction tokens during the mean pooling process, as these tokens have already influenced the output features through self-attention. Experimental results show that ID-0 achieves a 0.1%/0.3% improvement compared to ID-1 and 2.0%/12.1% improvement compared to ID-2 in Table 2. We hypothesize that this improvement arises primarily from that: the query inherently incorporates instruction information during forward propagation thought bidirectional self-attention mechanism. Therefore, by filtering out instruction tokens and focusing on feature comparisons between the query and candidate, the approach mitigates calculation bias. While the numerical improvement is modest, this step effectively eliminates theoretical instruction bias without compromising performance.

Table 2: Comparison of instruction integration

| ID | Instructions | Masking | Local Avg. | Global Avg. |
|----|------|------|------|------|
| 0 | ✓ | ✓ | 57.3 | 55.5 |
| 1 | ✓ | ✗ | 57.2 | 55.2 |
| 2 | ✗ | - | 55.3 | 43.4 |

Table 3: Comparison of progressive transition

| ID | Methods | Local Avg. | Global Avg. |
|----|------|------|------|
| 0 | Instruction Tuning on M-BEIR | 56.6 | 53.9 |
| 1 | [0] + text-only retrieval | 57.3 | 55.5 |
| 2 | [1] + text-image retrieval | 57.7 | 55.8 |

### 3.1.3 PROGRESSIVE TRANSITION

> **Finding 3**: Progressive transition effectively adapts decoder-only MLLMs to embedding models through stepwise training.

We set a baseline by vanilla instruction tuning on multimodal retrieval data from the M-BEIR dataset, presented in Table 3 under ID-0. Following prevalent approaches (Liu et al., 2024b; Jiang et al., 2024a), we use text-only retrieval data for pre-training, which improves performance on various retrieval tasks. Training on the NLI dataset (Gao et al., 2021) significantly enhances our model's performance, as indicated by ID-1 in Table 3. Despite these improvements, a notable gap persists when transitioning from single-modality, single-instruction tasks to multimodal, multi-instruction tasks. To address this gap, we propose a progressive approach incorporating an additional pre-training step using text-image retrieval data from the CC3M dataset (Sharma et al., 2018). This step further enhances performance, as demonstrated by ID-2 in Table 3. Since LLM was originally trained with causal attention, switching to bidirectional attention degrades text-visual encoder alignment. We speculate that text-based fine-tuning strengthens the text encoder's semantic representation, while image-text pairs further align the text-visual encoders. During experiments, we train the model with NLI using unidirectional InfoNCE, followed by CC3M using bidirectional InfoNCE. Furthermore, we further analyze the impact of training data selection in Appendix D, which reveals that the concise text in CC3M aligns better with retrieval tasks compared to other datasets (e.g., ShareGPT4V (Hurst et al., 2024) and TART (Asai et al., 2023)).

Therefore, we formally propose the progressive transition method, adopting a stepwise training strategy to adapt decoder-only MLLMs for embedding tasks. Following the principle of advancing from simpler to more complex tasks (Bengio et al., 2009), this phased approach ensures a smooth transition through three key steps: (1) Text Retrieval Adaptation: Learning retrieval tasks using text-only datasets to establish foundational capabilities. (2) Cross-modal Alignment: Advancing to cross-modal retrieval tasks through text-image paired data. (3) Instruction-tuned Multimodal Retrieval: Mastering instruction-guided multimodal retrieval tasks with comprehensive training on multimodal datasets.

### 3.2 HOW TO TRAIN MLLM-BASED EMBEDDERS BY INFONCE?

In this subsection, we investigate how MLLM-based embedding models should be trained under the contrastive learning framework, following the paradigm adopted by most existing methods. Despite its simplicity, we identify that several overlooked factors exert a critical impact on the final performance, including interactions among the training parameters of InfoNCE and the control of noise in hard negative mining during the continual training process.

### 3.2.1 INTERACTIONS AMONG LARGE BATCH SIZE, LEARNING RATE AND TEMPERATURE

> **Finding 4:** Increasing batch size yields performance gains, but these improvements plateau without appropriate learning rate scaling. Additionally, learnable temperature parameters play a pivotal role in enhancing the effectiveness of contrastive learning.

Our experimental results dispute the widely held belief that "increasing batch size directly improves performance", indicating that this assumption requires closer examination. By analyzing the results presented in Table 4, we identify three key findings as follows. (1) Experiments with ID-0, ID-2, and ID-4 indicate that increasing the batch size does improve model performance; however, the performance gains tend to plateau as the batch size continues to grow, as observed in comparisons

Table 4: Comparison of performance when increasing batch size

| ID | Batch Size | Temp ($\tau$) | LR ($\eta$) | Local Avg. |
|----|-----------|---------------|-------------|-----------|
| 0 | 480 | 0.05 (fixed) | $1 \times 10^{-4}$ | 57.2 |
| 1 | 1920 | 0.05 (fixed) | $1 \times 10^{-4}$ | 57.4 |
| 2 | 1920 | 0.05 (fixed) | $2 \times 10^{-4}$ | 58.3 |
| 3 | 3840 | 0.05 (fixed) | $2.8 \times 10^{-4}$ | 58.5 |
| 4 | 3840 | 0.05 (fixed) | $4 \times 10^{-4}$ | 58.9 |
| 5 | 5760 | 0.05 (fixed) | $6 \times 10^{-4}$ | 58.7 |
| 6 | 7680 | 0.05 (fixed) | $6 \times 10^{-4}$ | 58.9 |
| 7 | 3840 | 0.05 (learnable) | $4 \times 10^{-4}$ | 60.1 |
| 8 | 7680 | 0.05 (learnable) | $4 \times 10^{-4}$ | 60.3 |

between ID-5 and ID-6. (2) Consistent with prior research (Goyal et al., 2017) that highlights the necessity of scaling the learning rate (lr) alongside batch size, our contrastive learning experiments reveal similar trends. Specifically, simply increasing the batch size without adjusting the learning rate results in marginal improvements (ID-0 vs. ID-1). In contrast, applying scaling rule (ID-2) significantly boosts performance to 58.3%, which aligns with the findings from previous works. (3) Our investigation into the effects of temperature parameters reveals that employing a learnable temperature significantly enhances performance. The learnable mechanism effectively optimizes the sharpness of the probability distribution during training, superior to fixed temperature settings used in prior works such as LamRA (Liu et al., 2024b) and MM-Embed (Lin et al., 2024). Empirical results (comparing ID-4/ID-6 with ID-7/ID-8) demonstrate that this dynamic adjustment yields consistent performance gains (e.g., $+1.4\%$ in Local Avg), while also exhibiting stable and adaptive convergence across different batch sizes. More detailed ablation study on learnable temperature is provided in Appendix D.

### 3.2.2 CONTINUAL TRAINING WITH HARD NEGATIVE MINING

> **Finding 5:** Hard negatives may hinder convergence during training. Filtering false negatives and mixing random in-batch negatives help balance difficulty and improve performance.

The initial random negative sampling yields basic discriminative ability but struggles with challenging false positives: negatives that exhibit high semantic similarity yet are incorrect matches. Recent studies have employed hard negative mining (Lin et al., 2024; Lan et al., 2025) or hard negative reweighting in loss computation (Hou & Li, 2023; Radenovic et al., 2023; Zhuang et al., 2024) to address this issue. In this work, we explore related settings. However, as shown in Table 5 for ID-0, directly selecting the top-$k$ hard negatives proved to be excessively challenging, resulting in complete failure. Simply setting $k$ to a larger value (e.g., 50) prevents model collapse but fails to yield any performance improvement. Additionally, incorporating in-batch negatives results in a performance decline compared to the baseline in this table, as observed by ID-1. We hypothesize that some hard negatives are false negatives, which can adversely affect model training. Specifically, naively maximizing the loss on all hard negatives often leads to model collapse because the model is forced to push away semantically valid candidates that are labeled as negatives due to dataset noise. To address this issue, we propose a straightforward filtering method that eliminates hard negatives with scores exceeding a predefined threshold. This approach led to a significant improvement, as demonstrated by ID-2.

We apply a hard negative mining strategy as follows: (1) Feature Extraction: Using the model from the previous training stage, we extract features for all queries and candidates, compute similarity scores (excluding positive matches), and rank candidates to identify the most challenging negatives per query; (2) Filtered Hard Negative: For each query, we filter out negative samples that exceed a predefined threshold (considered as false negatives) and then select the top-$k$ negative samples as hard negatives; (3) Balanced Training: To prevent model convergence difficulties from excessive hard negatives and to accelerate training, we select a suitable $k$-value and mix these hard negatives with other in-batch negatives. We then perform continual finetuning by InfoNCE. In our experiments, we set $k = 5$ and the threshold to 0.7, filtering out any negative samples with scores exceeding this limit. The choice of $k = 5$ for hard negative mining strikes a balance between training efficiency and sample quality.

Table 5: Comparison of different strategies in hard negative mining

| ID | Methods | Local Avg. | Global Avg. |
|---|---|---|---|
| Progressive Transition (baseline) | in-batch neg | 60.6 | 58.7 |
| 0 | only top-$k$ hard neg | failed | failed |
| 1 | in-batch neg and top-$k$ hard neg | 57.4 | 55.4 |
| 2 | in-batch neg and filtered top-$k$ hard neg | 61.7 | 59.9 |

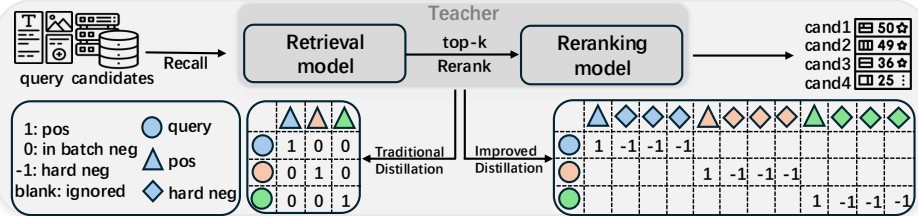

Figure 2: Distillation Illustration. It shows how the teacher model generates scores, and compares the traditional and improved distillation methods.

## 3.3 WILL RERANKER DISTILLATION IMPROVE PERFORMANCE?

> **Finding 6:** The improved distillation approach reduces computational overhead while increasing feature diversity, thereby enabling resource-efficient and effective distillation.

Some studies (Liu et al., 2024b; Lin et al., 2024) further refine retrieval results by applying a reranking step on top of the recall model. The reranker can be either a trained model (Liu et al., 2024b) or a zero-shot reranker (Lin et al., 2024) based on MLLMs. While this recall-then-rerank strategy often improves retrieval performance, it introduces additional inference latency and increases system complexity. In this work, we exploit to distill this cascaded pipeline into a single unified model, thereby simplifying the system while maintaining high performance.

Inspired by (Liu et al., 2024b), we train a generative reranker based on the decoder-only MLLM architecture. The training data is prepared by: (1) extracting embeddings for all queries and candidates via the recall model from Sec. 3.2.2, then (2) constructing pairs for each query consisting of a positive match and the top-50 most challenging negatives. The model is prompted to output "YES" for positives and "NO" for negatives, optimized with next-token prediction loss to maximize the likelihood of correct response. More detailed specifics can be found in Appendix C.

After training the reranker, we employ it to refine the ranking results from the recall model, further improving the retrieval performance. Specifically, the recall model from Sec. 3.2.2 first retrieves the top-$k$ candidates, then each (query, candidate) pair is processed by the reranker sequentially. The reranker is prompted to judge each candidate with a "YES" or "NO" response, where the probability of "YES" serves as the reranker's score. To further enhance overall performance, we combine the scores from both models through linear interpolation: $S_{multi} = \alpha \cdot S_{\text{recall}} + (1 - \alpha) \cdot S_{\text{rerank}}$, where $s_{multi}$ becomes the final relevance score, and $\alpha$ is a weight hyperparameter (fixed at 0.5). This completes our recall-then-rerank pipeline, producing optimized query-candidate matching scores.

We distill this entire pipeline as a teacher model into a single student model, shown in Figure 2, where the combined knowledge is compressed into one unified model by using KL divergence defined as Eq. 2.

$$\mathcal{L}_{\text{distill}} = D_{KL}(S_{multi} \parallel S_{single}) = \sum_i S_{multi}(i) \log \frac{S_{multi}(i)}{S_{single}(i)} \tag{2}$$

where $S_{multi}$ represents the multi-model ensemble scores and $S_{single}$ the single model's predictions. The scores are softmax-normalized before computing the KL divergence. In traditional distillation frameworks, the base model typically aligns with the reranker. Distillation samples are structured as (query, pos), with similarity matrices incorporating in-batch negatives. However, recall-then-rerank inference over this similarity score matrix incurs prohibitive computational costs, making it impractical for deployment.

We propose an improved distillation method to address this limitation. Specifically, we construct samples in the form of (query, positive, top-$k$ hard negatives), and confine distillation to the scope of positive and hard negative samples. In this way, it significantly reduces computational overhead while substantially increasing the diversity of features encountered during model training. Thus, it achieves both resource efficiency and performance gains—critically, it makes distillation practically feasible. Furthermore, besides the data and base model, a key factor contributing to the effectiveness of this distillation method lies in its tight alignment with the recall-then-rerank inference pipeline, depicted in Figure 2: (1) Retrieving with a hard negative model; (2) Reranking the top-$k$ results and generating integrated scores. Our distillation method mirrors this workflow: it adopts the hard negative model as the base and distills top-$k$ scores from the integrated model.

To further validate its efficiency and effectiveness, we provide the following mathematical analysis. Let $n$ denote the number of queries in a batch, and $N$ the total number of queries. In traditional distillation, the computational complexity of inferring the similarity matrix is $\frac{N}{n} \times (2n + n^2) = N(2+n)$, and the number of features observed per training epoch is $2N$. The total time for inference and training is $(6 + n)N$. In our improved method, the cost of computing the similarity matrix is reduced to $\frac{N}{n} \times (n(k + 3)) = N(k + 3)$, and the number of features observed during training increases to $(k + 2)N$. The total time is reduced to $(3k + 7)N$. Substituting our actual parameters, $n = 60 \times 64$ and $k = 50$, the improved method reduces the total time to $\frac{3 \times 50 + 7}{6 + 60 \times 64} = 4.1\%$ of the original, while the number of features seen during training increases by $26.0\times$. In terms of actual training time, improved distillation completes in just 14 hours, compared to a theoretical cost of over 340 hours for the traditional method. This substantial efficiency gain makes MLLM distillation practically feasible, transforming it from an intractable task to a manageable one. More detailed information can be found in Appendix E and Table 14.

Table 6: Comparison of distillation from rerankers

| Methods | Local Avg. | Global Avg. |
|---|---|---|
| Hard Negative Mining (baseline) | 61.7 | 59.9 |
| Recall-Reranker | 64.5 | 61.7 |
| Distillation | 63.2 | 60.7 |
| Continue-hard | 62.2 | 60.0 |

As shown in Table 6, the "Recall-Reranker" represents the model we aim to distill, and the "Distillation" method achieves an improvement of 1.5%/0.8% over the baseline. However, distillation data are generated from the top-$k$ retrieval results from baseline model and correspond to hard negative samples for it. We designed experiment "Continue-hard" for validation. Unlike the "Distillation" stage, the "Continue-hard" model continues training by treating the training data as hard negatives. The "Continue-hard" method demonstrates an improvement of 0.5%/0.1% over the baseline; however, it exhibits a decline of 1.0%/0.7% when compared to the "Distillation" approach. It validates that the performance improvement is clearly attributable to the distillation process. Notably, contrary to the instability observed in Section 3.2.2 with the "only top-$k$ hard negatives" setting, the "Continue-hard" strategy ($k = 50$) employed here remains stable and achieves slight gains. This stability is primarily attributed to the strong initialization: starting from a converged Recall model provides robust feature representations that are more resilient to hard negatives. Additionally, the larger $k$ value helps mitigate the noise introduced by false negatives.

## 3.4 U-MARVEL FRAMEWORK

Motivated by the findings presented above, we introduce a unified framework, term U-MARVEL (**U**niversal **M**ultimod**A**l **R**etrie**V**al via **E**mbedding **L**earning), which consists of the following three stages: (1) Progressive transition: the model is progressively fine-tuned on the increasing complexity levels of the retrieval data, which enables the model to gradually adapt to retrieval tasks. (2) Hard Negative Mining and Fusion Reranker Model: building on Progressive transition, we first train a hard negative model for recall and a reranker model for ranking, and then linearly combine them to obtain a recall–reranker model. (3) Distillation: we perform improved knowledge distillation on the recall–reranker model. Detailed experimental settings and results for U-MARVEL are provided in Table 12 in Appendix C and Table 15 in Appendix D, respectively. Additionally, to demonstrate the generalization ability of our framework across different model sizes, we also provide results using the Qwen3-VL-4B-Instruct (Team, 2025) backbone in Table 16 and Appendix D.

# 4 COMPARISONS WITH STATE-OF-THE-ART METHODS

In this section, we compare U-MARVEL with state-of-the-art methods in both supervised and unsupervised settings.

## 4.1 M-BEIR EVALUATION

We first compare U-MARVEL with competitors on M-BEIR benchmark (Wei et al., 2024) in both local and global pool settings, details of this dataset can be found in Appendix A. For fair comparison, we also present results of the recall-then-rerank cascade pipeline, denoted as U-MARVEL$^+$, similar to that used in LamRA (Liu et al., 2024b), details seen in Appendix C. As shown in Table 7 and 17, U-MARVEL establishes a new state-of-the-art. Beyond the raw metrics, three key observations highlight the efficacy of our design: 1) **Efficiency-performance trade-off**: In the single-model setting, U-MARVEL significantly outperforms existing state-of-the-art approaches by a large margin. Furthermore, while comparisons with stronger baselines like LamRA (Liu et al., 2024b) typically involve a computationally expensive two-stage retrieval process, U-MARVEL achieves comparable performance using only single-stage inference. This dual advantage of superior accuracy and efficiency is directly attributable to our improved distillation strategy (Sec. 3.3), which successfully transfers the discriminative power of the reranker into the embedding space. 2) **Robustness across settings**: As detailed in Table 17 and Appendix D, our method maintains its superiority even in the more challenging "Global Pool" setting, where candidates from all tasks are mixed. It suggests that our method helps the model learn more robust, task-agnostic features rather than overfitting to specific task distributions. 3) **Contribution of Components**: The ablation study in Table 15 (Appendix C) further reveals that our model already surpasses existing state-of-the-art methods in the single-model setting upon the completion of "Progressive Transition". Subsequently, "Hard Negative Mining" provides an additional boost in discrimination, while the "Distillation" stage is critical for closing the gap with reranking models.

Table 7: Comparisons with SoTA approaches on M-BEIR benchmark in local pool setting.

| Methods | $q^t \to c^i$ | | | $q^t \to c^t$ | | $q^t \to (c^i, c^t)$ | $q^i \to c^t$ | | | | $q^i \to c^i$ | | $(q^i, q^t) \to c^i$ | | $(q^i, q^t) \to c^t$ | | Avg. |
|---|---|---|---|---|---|---|---|---|---|---|---|---|---|---|---|---|---|
| | VN | CO | F200 | WQ | ES | WQ | VN | CO | F200 | NS | ON | InS | FQ | CR | ON | InS | |
| | R@5 | R@5 | R@10 | R@5 | R@5 | R@5 | R@5 | R@5 | R@10 | R@5 | R@5 | R@5 | R@10 | R@5 | R@5 | R@5 | |
| *Single model* | | | | | | | | | | | | | | | | | |
| CLIP-L (Radford et al., 2021) | 43.3 | 61.1 | 6.6 | 36.2 | 43.3 | 45.1 | 41.3 | 79 | 7.7 | 26.1 | 24.2 | 20.5 | 7 | 13.2 | 38.8 | 26.4 | 32.5 |
| SigLIP (Zhai et al., 2023) | 30.1 | 75.7 | **36.5** | 39.8 | 27 | 43.5 | 30.8 | 88.2 | 34.2 | 28.9 | 29.7 | 25.1 | 14.4 | 22.7 | 41.7 | 27.4 | 37.2 |
| UniIR-BLIP (Wei et al., 2024) | 23.4 | 79.7 | 26.1 | 80 | 50.9 | 79.8 | 22.8 | 89.9 | 28.9 | 33 | 41 | 22.4 | 29.2 | 52.2 | 55.8 | 33 | 46.8 |
| UniIR-CLIP (Wei et al., 2024) | 42.6 | 81.1 | 18 | 84.7 | 59.4 | 78.7 | 43.1 | 92.3 | 18.3 | 32 | 45.5 | 27.9 | 24.4 | 44.6 | 67.6 | 48.9 | 50.6 |
| LamRA-Ret (Liu et al., 2024b) | 41.6 | 81.5 | 28.7 | 86 | 62.6 | 81.2 | 39.6 | 90.6 | 30.4 | 32.1 | 54.1 | 52.1 | 33.2 | 53.1 | 76.2 | 63.3 | 56.6 |
| U-MARVEL | **47.3** | **84.4** | 33.6 | **97.1** | **78.8** | **88.5** | **47.3** | **93.5** | **35.1** | **34.2** | **62.5** | **58.3** | **36.4** | **60.7** | **79.4** | **74.7** | **63.2** |
| *+Reranker* | | | | | | | | | | | | | | | | | |
| LamRA (Liu et al., 2024b) | 48 | 85.2 | 32.9 | 96.7 | 75.8 | 87.7 | 48.6 | 92.3 | 36.1 | 33.5 | 59.2 | **64.1** | 37.8 | **63.3** | 79.2 | 78.3 | 63.7 |
| U-MARVEL$^+$ | **49.4** | **85.6** | 34.2 | **98.5** | **81.4** | **89.4** | **50.5** | 88.4 | **37.7** | **34.7** | **63.7** | 62.9 | **38.2** | 63.2 | **80.8** | **78.9** | **64.8** |

## 4.2 ZERO-SHOT EVALUATION

We evaluate the performance of U-MARVEL with several unseen datasets in a zero-shot manner. Details regarding the datasets and evaluation metrics are provided in Appendix B. As presented in Table 8, U-MARVEL achieves state-of-the-art performance on 9 out of 12 tasks, highlighting its exceptional zero-shot capabilities compared to leading methods such as VLM2Vec (Jiang et al., 2024b), UniME (Gu et al., 2025), and LamRA (Liu et al., 2024b). This verifies that our model effectively transfers knowledge from the diverse M-BEIR tasks to unseen domains, likely due to the high diversity of the progressive training curriculum that prevents catastrophic forgetting of the pre-trained knowledge. Furthermore, we evaluate U-MARVEL on the zero-shot text-to-video retrieval task. The results on the MSVD and MSR-VTT datasets, summarized in Table 9, demonstrate that U-MARVEL outperforms VLM2Vec (Jiang et al., 2024b), LamRA (Liu et al., 2024b), and LLaVE-7B (Lan et al., 2025). However, it is worth noting that U-MARVEL$^+$ exhibits degraded performance, which may be attributed to fail to model temporal contexts, such as action sequence progression.

Table 8: Comparisons with SoTA approaches on zero-shot image and text benchmarks.

| Methods | $q^i \to c^t$ | | | $q^t \to c^i$ | | | $(q^i, q^t) \to c^i$ | | $q^{dialog} \to c^i$ | $(q^i \oplus q^t) \to c^i$ | ITM | |
|---|---|---|---|---|---|---|---|---|---|---|---|---|
| | ST4V R@1 | U-1K* R@1 | Flickr R@1 | ST4V R@1 | U-1K* R@1 | Flickr R@1 | CIRCO* MAP@5 | GCIS* R@1 | V-Dia* R@1 | M-FIQ* R@5 | C-Neg Acc. | S-Cre* Acc. |
| *Single model* | | | | | | | | | | | | |
| CLIP-L (Radford et al., 2021) | 84 | 52.8 | 67.3 | 81.8 | 68.7 | 87.2 | 4 | 13.3 | 23.7 | 17.7 | 66.7 | 73 |
| UniIR-CLIP$_{SF}$ (Wei et al., 2024) | 85.8 | 75 | 78.7 | 84.1 | 78.4 | **94.2** | 12.5 | 16.8 | 26.8 | 39.4 | 79.9 | 80.3 |
| E5-V (Jiang et al., 2024a) | 86.7 | 84 | 79.5 | 84 | 82.4 | 88.2 | 24.8 | 18.5 | 54.6 | 19.2 | 83.2 | 84.7 |
| MagicLens-L (Zhang et al., 2024b) | 85.5 | 59.3 | 72.5 | 60.9 | 24.2 | 84.6 | 29.6 | 16.3 | 28 | 22.6 | 62.7 | 75.9 |
| VLM2Vec (Jiang et al., 2024b) | 90.7 | 90.8 | 76 | 85.8 | 84.7 | 90.6 | - | - | - | - | - | 79.5 |
| UniME (Gu et al., 2025) | **97.2** | 95.9 | 81.9 | 93.9 | **95.2** | 93.4 | - | - | - | - | - | 85 |
| LamRA-Ret (Liu et al., 2024b) | 93.3 | 95.1 | 82.8 | 88.1 | 94.3 | 92.7 | 33.2 | 18.9 | 62.8 | 60.9 | 79.6 | 85.8 |
| U-MARVEL | 96.4 | **96.7** | **85.4** | **97.2** | 93.5 | 93.3 | **36.2** | **19.1** | **70.3** | **65.7** | **84.5** | **87.9** |
| *+Reranker* | | | | | | | | | | | | |
| LamRA (Liu et al., 2024b) | **97.9** | **98.8** | 88.1 | 96.5 | 98 | **97.6** | 42.8 | **24.8** | 70.9 | 63.9 | 85.9 | **93.5** |
| U-MARVEL$^+$ | 97.8 | 97.7 | **88.5** | **98.9** | **98.2** | 95.1 | **46.0** | 22.6 | **77.6** | **66.3** | **86.1** | 93.4 |

Table 9: Comparisons with SoTA approaches on zero-shot text-to-video retrieval benchmarks.

| Model | MSR-VTT | | | MSVD | | |
|---|---|---|---|---|---|---|
| | R@1 | R@5 | R@10 | R@1 | R@5 | R@10 |
| *Zero-shot (finetuned with text-video data)* | | | | | | |
| InternVideo (Wang et al., 2022) | 40.0 | 65.3 | 74.1 | 43.4 | 69.9 | 79.1 |
| ViCLIP (Wang et al., 2023) | 42.4 | - | - | 49.1 | - | - |
| UMT-L (Li et al., 2023b) | 42.6 | 64.4 | 73.1 | 49.9 | 77.7 | 85.3 |
| InternVideo2$_{s2}$-6B (Wang et al., 2024b) | 55.9 | 78.3 | 85.1 | 59.3 | 84.4 | 89.6 |
| *Zero-shot (finetuned only with text-image data)* | | | | | | |
| VLM2Vec (Jiang et al., 2024b) | 43.5 | 69.3 | 78.9 | 49.5 | 77.5 | 85.7 |
| LamRA (Liu et al., 2024b) | 44.7 | 68.6 | 78.6 | 52.4 | 79.8 | 87.0 |
| LLaVE-7B (Lan et al., 2025) | 46.8 | 71.1 | 80.0 | 52.9 | 80.1 | 87.0 |
| U-MARVEL | **47.2** | **72.0** | **80.2** | **54.6** | **80.9** | **87.7** |
| U-MARVEL$^+$ | 47.5 | 69.7 | 78.5 | 53.1 | 79.9 | 87.1 |

# 5 CONCLUSION AND LIMITATIONS

This paper systematically explores the design principles for building high-performing universal multimodal retrievers with MLLMs. Through a comprehensive study, we identified key factors that significantly impact retrieval performance, including embedding generation strategies, progressive adaptation techniques, and training recipes such as hard negative mining and re-ranker distillation. Based on these insights, we introduce U-MARVEL, a unified framework that achieves state-of-the-art performance on the M-BEIR benchmark in supervised settings and demonstrates strong zero-shot generalization across diverse tasks. The results highlight the generalization potential of U-MARVEL and its ability to address the complex requirements of real-world retrieval scenarios. We believe that our study contributes valuable knowledge to the research community and paves the way for future advancements in universal multimodal retrieval.

Despite the promising results achieved by U-MARVEL, there remain several limitations that can be improved. First, it focuses on text and image modalities, leaving the inclusion of other modalities (e.g., audio) for future work. Second, while effective in zero-shot scenarios, its integration with LLMs in retrieval-augmented generation (RAG) applications remains underexplored. Lastly, experiments were limited to 7B-sized models due to submission time constraints; future work will extend support models of various sizes, enabling flexible trade-offs between performance and cost.

## ETHICS STATEMENT

This study strictly adheres to the full requirements of the Code of Ethics outlined by the International Conference on Learning Representations (ICLR) . We have thoroughly read and adhered to this code during the research design, data handling, experimental execution, and manuscript writing phases, ensuring that the entire research process aligns with academic integrity and ethical guidelines.

REPRODUCIBILITY STATEMENT

To ensure the reproducibility of this study, we fixed the global random seed to 42 across Python, NumPy, and PyTorch environments. This configuration minimizes stochasticity in model initialization, data sampling, and training dynamics. Consequently, the results reported in Tables 1–9 and Tables 13–18 reflect deterministic runs, aligning with standard practices in large-scale model research where computational constraints render multi-seed training infeasible. To facilitate direct replication, the source code and datasets are publicly available via the GitHub repository linked in the Abstract. Additionally, detailed experimental environment configurations are included in the Appendix.

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

APPENDIX

## A  M-BEIR DATASET & METRICS

**Datasets.**  We evaluate the retrieval performance of our method with the M-BEIR (Wei et al., 2024) benchmark. This dataset consist of 8 possible multimodal retrieval tasks involving texts and images. It is constructed based on 10 different datasets from 4 domains. Moreover, according to the datasets and task types (i.e., the modalities of queries and candidates), the multi-modal retrieval tasks are further subdivided into 16 types.

Regarding the structure of the dataset, the M-BEIR consists of three key components: instructions, query sets, and candidate pools. Each task has its independent query set and candidate pool, and the query set and candidate pool of each task contain only a single modality. Additionally, each task is equipped with 4 instructions, which clearly specify the target modality of the task. Regarding the scale of the dataset, the training set contains 1.33 million queries and 1.93 million candidates, while the test set contains 0.19 million queries and 5.6 million candidates. In addition, besides the local query sets and candidate pools for each task, both the training set and the test set also have global query sets and global candidate pool, that is, the query sets and candidate pools of the 16 tasks are combined. Table 10 provides a detailed presentation of the information about the M-BEIR dataset.

Table 10: The overview of M-BEIR training/validation/test set.

| Task | Dataset | Instruction (shown 1 out of 4) | Domain | Train | Dev | Test | Pool |
|---|---|---|---|---|---|---|---|
| 1. $q^t \rightarrow c^i$ | VisualNews (Liu et al., 2020) | Identify news-related image match with the description | News | 99K | 20K | 20K | 542K |
| | MSCOCO (Lin et al., 2014) | Find an everyday image match with caption | Misc. | 100K | 24.8K | 24.8K | 5K |
| | Fashion200K (Han et al., 2017) | Based on fashion description, retrieve matched image | Fashion | 15K | 1.7K | 1.7K | 201K |
| 2. $q^t \rightarrow c^t$ | WebQA (Chang et al., 2022) | Find an paragraph from Wikipedia to answer the question | Wiki | 16K | 1.7K | 2.4K | 544K |
| 3. $q^t \rightarrow (c^i, c^t)$ | EDIS (Liu et al., 2023b) | Find a news image matching with the caption | News | 26K | 3.2K | 3.2K | 1M |
| | WebQA (Chang et al., 2022) | Find a Wiki image that answer the question | Wiki | 17K | 1.7K | 2.5K | 403K |
| 4. $q^i \rightarrow c^t$ | VisualNews (Liu et al., 2020) | Provide a news-related caption for the displayed image | News | 100K | 20K | 20K | 537K |
| | MSCOCO (Lin et al., 2014) | Find a caption describe the an image | Misc. | 113K | 5K | 5K | 25K |
| | Fashion200K (Han et al., 2017) | Find a description for the fashion item in the image | Fashion | 15K | 4.8K | 4.8K | 61K |
| 5. $q^i \rightarrow c^i$ | NIGHTS (Fu et al., 2023) | Find an image that is identical to the given image | Misc. | 16K | 2K | 2K | 40K |
| 6. $(q^i, q^t) \rightarrow c^t$ | OVEN (Hu et al., 2023) | Retrieve a Wiki text that answer the given query about the image | Wiki | 150K | 50K | 50K | 676K |
| | InfoSeek (Chen et al., 2023) | Find an article that answers the given question about the image | Wiki | 141K | 11K | 11K | 611K |
| 7. $(q^i, q^t) \rightarrow c^i$ | FashionIQ (Wu et al., 2021) | Find an image to match the fashion image and style note | Fashion | 16K | 2K | 6K | 74K |
| | CIRR (Liu et al., 2021) | I'm looking for a similar everyday image with the described changes | Misc. | 26K | 2K | 4K | 21K |
| 8. $(q^i, q^t) \rightarrow (c^i, c^t)$ | OVEN (Hu et al., 2023) | Find a Wiki image-text pair to answer a question regarding an image | Wiki | 157K | 14.7K | 14.7K | 335K |
| | InfoSeek (Chen et al., 2023) | Find a Wiki image-text pair to answers my question about this image | Wiki | 143K | 17.6K | 17.6K | 481K |
| | 10 datasets | 64 instructions | 4 domains | 1.1M | 182K | 190K | 5.6M |

**Metrics.**  Following the evaluation protocols by M-BEIR benchmark, we use the Recall@10 (R@10) for Fashion200K (Han et al., 2017) and FashionIQ (Wu et al., 2021), and the Recall@5 (R@5) for the remaining datasets. We report the average score of all test queries as the criterion for measuring the retrieval accuracy.

The evaluation has two kinds of settings: **Local** and **Global** pool settings. The local setting indicates that each task conducts retrieval operations within its own candidate pool and the retrieved results will not have modality errors. The global setting means that the retrieval is carried out within the global candidate pool formed by combining all task candidate pools. In this case, modality errors will occur in the retrieved results.

Note that the first columns of Table 7, 15 and 10 indicates the retrieval task type: $q^t$ for text queries, $q^i$ for image queries, $c^t$ for text candidates, and $c^i$ for image candidates.

Datasets used in our experiments include VisualNews (VN), MSCOCO (CO), Fashion200K (F200), WebQA (WQ), EDIS (ES), NIGHTS (NS), OVEN (ON), InfoSeek (InS), FashionIQ (FQ), and CIRR (CR) in Table 7 and 17.

## B  ZERO-SHOT DATASETS & METRICS

**Datasets.**  We have examined the zero-shot capabilities of U-MARVEL in retrieval tasks, specifically focusing on image-text cross-modal retrieval and text-video retrieval. For tasks of cross-modal retrieval between images and texts, we use the following datasets: ShareGPT4V (Hurst et al.,

2024), Urban-1K (Zhang et al., 2024a), CIRCO (Baldrati et al., 2023), Flickr (Plummer et al., 2015), GeneCIS (Vaze et al., 2023), Visual Dialog (Das et al., 2017), Multi-round FashionIQ (Yuan & Lam, 2021), CC-Neg (Singh et al., 2024), and Sugar-Crepe (Hsieh et al., 2023). For text-video retrieval, the datasets used are MSR-VTT (Xu et al., 2016) and MSVD (Chen & Dolan, 2011). We present the details of the Unseen Dataset in Table 11. Although some of them are actually adapted from MSCOCO or FashionIQ, we still treat these datasets as unseen datasets due to the significant differences in their captions or query formats, following (Liu et al., 2024b). For instance, the captions in Urban1K consist of extended captions generated by GPT-4V (Hurst et al., 2024), while the query format of CIRCO combines a reference image with a relative caption. These differences create a substantial disparity compared to the original COCO (Lin et al., 2014) dataset. Table 11 provides a detailed presentation of the information about the zero-shot datasets.

**Metrics.** The first row of Table 8 indicates the retrieval task type: $q^t$ for text queries, $q^i$ for image queries, $q^{dialog}$ for dialog queries, $(q^i \oplus q^t)$ for multi-interleaved image-text queries, $c^t$ for text candidates, $c^i$ for image candidates, and ITM for the Image-Text Matching task. Datasets used in our experiments include ShareGPT4V (ST4V), Urban-1K* (U-1K*), Flickr, CIRCO* (CIRCO*), GeneCIS* (GCIS*), Visual Dialog* (V-Dia*), Multi-round FashionIQ* (M-FIQ*), CC-Neg (C-Neg), and Sugar-Crepe* (S-Cre*) in Table 8. * indicates that the images in these datasets are sourced from COCO or FashionIQ.

Table 11: Summary of the evaluation benchmarks. # Queries represents the number of test queries, and # Candidates denotes the number of test candidates per query.

| Benchmark | Zero-shot | # Queries | # Candidates |
|---|---|---|---|
| ShareGPT4V (Hurst et al., 2024) | ✓ | 1K | 1K |
| Urban-1K (Zhang et al., 2024a) | ✓ | 1K | 1K |
| Flickr30K (Plummer et al., 2015) | ✓ | 1K | 5K |
| CIRCO (Baldrati et al., 2023) | ✓ | 800 | 120K |
| GeneCIS (Vaze et al., 2023) | ✓ | 8K | 10 - 15 |
| Visual Dialog (Das et al., 2017) | ✓ | 2K | 2K |
| Multi-round FashionIQ (Yuan & Lam, 2021) | ✓ | 2.4K | 6.2K |
| CC-Neg (Singh et al., 2024) | ✓ | 40K | 2 |
| Sugar-Crepe (Hsieh et al., 2023) | ✓ | 7.5K | 2 |
| MSR-VTT (Xu et al., 2016) | ✓ | 670 | 27K |
| MSVD (Chen & Dolan, 2011) | ✓ | 1K | 1K |

## C IMPLEMENTATION DETAILS

We select Qwen2-VL-7B-Instruct (Wang et al., 2024a) as the pretrained MLLM backbone to instantiate both of the retriever and the reranker. Throughout all experiments, we froze the visual side and employed LoRA (Hu et al., 2022) to fine-tune the LLM component of the model. For embedding models, the InfoNCE loss (Oord et al., 2018) is selected as our training objective:

$$\mathcal{L}_{\text{InfoNCE}} = - \log \frac{\exp(\text{sim}(e_q, e_c^+)/\tau)}{\sum_i \exp(\text{sim}(e_q, e_{c^i})/\tau)} \tag{3}$$

where $e_q$, $e_c^+$, $e_{c^i}$ represent the embeddings of the query, the positive candidate, and any sample from the candidate set, respectively. $\tau$ is the temperature parameter. $\text{sim}(\cdot, \cdot)$ means the cosine similarity.

**Recall-Rerank Model.** Some studies (Liu et al., 2024b; Lin et al., 2024) further refine retrieval results by applying a reranking step on top of the recall model. Inspired by (Liu et al., 2024b), we randomly sample top-$k$ hard negatives from the training hard negative stage to construct data triplets (query, positive, hard negative). Based on the progressive transition model, we perform supervised fine-tuning of the model shown in Figure 1(b) using the loss function defined in Eq. 4. Detailed training configuration are shown in Table 12.

$$\mathcal{L}_{\text{rerank}}(\theta) = - \log P_\theta(y \mid \text{query} + \text{candidate}) \tag{4}$$

where $y = \text{yes}$ if the candidate belongs to the set of positive candidates, and $y = \text{no}$ otherwise.

$$\text{score} = P_\theta(\text{YES} \mid \text{query} + \text{candidate}) \tag{5}$$

During inference, we adopt the reranker's predicted probability of the token "YES" as the relevance score, as shown in Eq. 5. After training the reranker, we employ the hard negative mining model for recall, obtaining the top-$k$ candidates $(c_1, c_2, \ldots, c_k)$ along with their corresponding retrieval scores $S_{\text{recall}} = (s_1, s_2, \ldots, s_k)$ for each query. These candidates are then evaluated using the reranker model, which produces scores $S_{\text{rerank}} = (s'_1, s'_2, \ldots, s'_k)$. The final ranking is generated by combining both sets of scores through Eq. 6. This completes our recall-then-rerank pipeline, producing optimized query-candidate matching scores.

$$S_{multi} = \alpha \cdot S_{\text{recall}} + (1 - \alpha) \cdot S_{\text{rerank},} \tag{6}$$

where $\alpha$ is set to 0.5.

**Improved Distillation.** For the distillation stage, we use the following loss function:

$$\mathcal{L}_{\text{distill}} = D_{KL}(S_{multi} \parallel S_{single}) = \sum_i S_{multi}(i) \log \frac{S_{multi}(i)}{S_{single}(i)} \tag{7}$$

where $S_{multi}$ represents the multi-model ensemble scores and $S_{single}$ the single model's predictions. The scores are softmax-normalized before computing the KL divergence.

Table 12: Training Configurations

| Training Stage | Temp ($\tau$) | LR ($\eta$) | Batch Size | GPU number | LoRA para. | epoch number |
|---|---|---|---|---|---|---|
| Progressive Transition (stage-1) | 0.05 (fixed) | $2 \times 10^{-4}$ | 576 | 8 | rank(64)$\alpha$(128) | 2 |
| Progressive Transition (stage-2) | 0.05 (fixed) | $1 \times 10^{-4}$ | 720 | 16 | rank(128)$\alpha$(256) | 1 |
| Progressive Transition (stage-3) | 0.05→0.0206* | $4 \times 10^{-4}$ | 3840 | 64 | rank(128)$\alpha$(256) | 1 |
| Hard-neg mining | 0.0206→0.0188* | $1.5 \times 10^{-5}$ | 1408 | 64 | rank(128)$\alpha$(256) | 1 |
| Distillation | 0.0188→0.0204* | $1 \times 10^{-5}$ | 50 | 64 | rank(128)$\alpha$(256) | 1 |
| Reranker Training | - | $2 \times 10^{-5}$ | 128 | 64 | rank(128)$\alpha$(256) | 1 |

*Learnable temperature parameter

**U-MARVEL$^+$.** We use the U-MARVEL model as the initial retrieval system and employ its Top-100 candidate results to train a point-wise reranking model following the methodology presented in Figure 1(b) and Eq. 4 as the loss function. Subsequently, the embedding similarity scores produced by the U-MARVEL model are combined with the reranking scores from the point-wise reranking model through a weighted fusion mechanism to compute the final ranking score:

$$S = \alpha \cdot S_{\text{U-MARVEL}} + (1 - \alpha) \cdot S_{\text{point-wise reranking model}}, \tag{8}$$

where $\alpha$ is a weighting hyperparameter, set to 0.5. Based on this final score, the candidate set is reranked to generate the final ranked list. We refer to this unified framework as U-MARVEL$^+$. The detailed training configurations are provided in Table 12.

# D SUPPLEMENTAL EXPERIMENTAL RESULTS

**Effectiveness of Three Key Stages in U-MARVEL.** We present the retrieval performance of each stage in U-MARVEL in Table 15, which provides a detailed ablation study of our framework. The results clearly demonstrate the effectiveness of the proposed design, showcasing significant performance improvements achieved through the progressive transition mechanism, hard negative mining, and reranker distillation. These components collectively contribute to the robustness and efficiency of U-MARVEL, highlighting the importance of each stage in enhancing retrieval capabilities and validating the overall framework design.

**Comparison of M-BEIR (global).** Table 17 presents a comprehensive comparison of our proposed U-MARVEL framework with state-of-the-art methods on the M-BEIR benchmark under the global pooling setting. The results demonstrate that, within the single-model category, U-MARVEL significantly outperforms leading approaches such as MM-Embed (Lin et al., 2024) and LamRA-Ret (Liu et al., 2024b), by a large margin. Furthermore, when integrated into a recall-then-rerank pipeline, the enhanced variant U-MARVEL$^+$ achieves the best overall performance, further solidifying the effectiveness of our framework. These findings highlight the versatility and superiority of U-MARVEL across different retrieval paradigms, showcasing its potential for advancing state-of-the-art retrieval systems.

**Held-out Experiments on M-BEIR.**   To evaluate the cross-task generalization ability of the proposed method, we trained the model on the five tasks, reserving the remaining three tasks as held-out tasks for evaluation. As presented in Table 18, our method achieves an average improvement of 2.4%/1.6% across the three held-out tasks compared with SoTA approaches.

**Analysis on Training Data for Progressive Transition.**   Our method adopts a progressive training pipeline: NLI → CC3M → M-BEIR, achieving scores of 57.7%/55.8% as shown in Table 3. During the progressive training phase, we attempted to incorporate high-quality image-text pairs from ShareGPT4V, a dataset characterized by highly detailed image captions. However, contrary to expectations, utilizing ShareGPT4V data—either independently or mixed with CC3M—resulted in performance degradation on the M-BEIR benchmark, as demonstrated in Table 13. We hypothesize that this decline stems from a significant distribution shift: the dense, exhaustive descriptive text in ShareGPT4V contrasts sharply with the concise, query-oriented text typical of retrieval tasks (e.g., M-BEIR). This discrepancy likely hinders the model's alignment with the target retrieval objective. Furthermore, we explored an alternative progressive path: NLI → TART (Asai et al., 2023) → M-BEIR, which also led to a substantial decrease in performance. We conjecture that the diverse text instruction tasks within TART interfere with the model's effective alignment for multimodal retrieval.

**Analysis on Learnable Temperature of InfoNCE.**   We analyze the impact of the learnable temperature parameter $\tau$, initialized at 0.05 in Progressive Transition (Stage-3).

- **Superior Performance over Fixed $\tau$:** Learnable $\tau$ consistently outperforms static settings. In Table 4, ID-7 (batch size 3840) and ID-8 (batch size 7680) achieve scores of 60.1% and 60.3%, respectively, surpassing the fixed $\tau = 0.05$ baseline (ID-4 and ID-6, both at 58.9%). We further conducted an experiment fixing $\tau = 0.02$ (aligned with the converged value of ID-7), which yielded a suboptimal score of 59.8%. This comparison confirms that dynamic adjustment adapts better to the optimization landscape than even a carefully manually tuned static value.

- **Adaptive Convergence:** We observe stable convergence of $\tau$ across different batch sizes. Specifically, we observed convergence to 0.0206 for batch size 3840 and 0.0195 for batch size 7680. The slight decrease in the larger batch setting aligns with findings in (Sun & Li, 2024), where a sharper distribution is necessitated to handle increased batch noise.

- **Stage-wise Evolution:** As detailed in Table 12, $\tau$ fluctuates adaptively across stages (Stage-3: 0.0206 → Hard-neg: 0.0188 → Distillation: 0.0204). Notably, $\tau$ drops during Hard-negative Mining to increase sensitivity to hard gradients.

**Analysis on Generalization Ability of U-MARVEL.**   To investigate the generalization ability of U-MARVEL across different MLLM backbones, we extended our framework to the Qwen3-VL-4B model. As presented in Table 16, the experimental results align consistently with the conclusions drawn from the 7B model, showing steady performance improvements across the three training stages—Progressive Transition, Hard Negative Mining, and Distillation. This validates that our proposed training recipe is robust and agnostic to specific model architectures. Remarkably, despite having significantly fewer parameters, the Qwen3-VL-4B model trained with U-MARVEL achieves a Local Average Recall of 58.8% and a Global Average Recall of 56.2% on the M-BEIR benchmark. These results not only demonstrate the effectiveness of our method but also establish a new state-of-the-art, surpassing the previous best-performing method, LamRA-Ret (Liu et al., 2024b) (Local: 56.6%, Global: 54.9%) with a larger 7B backbone. This underscores the strong generalization potential of U-MARVEL in enabling smaller models to achieve superior retrieval capabilities in both local and global search scenarios.

Table 13: Comparison of Progressive Transition with Different Training Data

| ID | Training Pipeline | Local Avg. | Global Avg. |
|---|---|---|---|
| 0 | NLI →ShareGPT4V→ M-BEIR | 56.9 | 54.0 |
| 1 | NLI → CC3M&ShareGPT4V → M-BEIR | 57.2 | 55.2 |
| 2 | NLI → TART → M-BEIR | 50.3 | 36.6 |

# E    DISTILLATION COMPUTATIONAL COST ANALYSIS

The distillation process consists of two main parts: (1) The teacher model infers a similarity matrix; (2) The student model is trained to extract features, compute scores, and learn from the teacher model's similarity matrix. In terms of computational cost, the recall stage primarily focuses on extracting the feature representations of the samples. On the other hand, the rerank stage is primarily concerned with calculating the similarity scores between the query and candidates. Despite the differences in tasks, the inference time for a single pass through both stages is approximately the same. Moreover, we assume that during the distillation phase, the training time for a feature by the student model is approximately twice that of inference, as it includes both forward and backward propagation. Table 14 presents a detailed comparison between the traditional and improved distillation methods. We adopt the following notation for our analysis:

- $N$: The total number of queries in the dataset (assuming a ratio of 1:1 with positive samples, the candidate pool size is also $N$).
- $n$: The number of queries in a single batch.
- $k$: The number of top-$k$ hard negatives selected in our improved method.

## E.1    TEACHER MODEL INFERENCE STAGE

This stage is subdivided into Recall and Rerank phases.

**Recall:**    Both the traditional and improved methods require feature extraction for all queries and candidate documents to compute retrieval scores. With $N$ queries and $N$ positive candidates, the total inference cost is $N + N = 2N$. For the improved method, the additional overhead of nearest neighbor search to identify top-$k$ hard negatives is computationally negligible compared to feature extraction and is therefore omitted.

**Rerank:**    This phase focuses on computing query-candidate similarity scores.

- Traditional Method: Computes scores for all pairs within a batch (utilizing in-batch negatives), resulting in an $n \times n$ similarity matrix per batch. The total cost is $\frac{N}{n} \times n^2 = nN$.
- Improved Method: Computes scores strictly for the query, the positive sample, and the mined hard negatives, resulting in an $n \times (k + 1)$ matrix per batch. The total cost is $\frac{N}{n} \times n(k + 1) = (k + 1)N$.

## E.2    STUDENT MODEL TRAINING STAGE

- Traditional Method: The student model processes the query and positive sample pairs, totaling $2N$ features. Applying the training cost multiplier (factor of 2 for forward/backward passes), the total training cost is $2 \times 2N = 4N$.
- Improved Method: The input augments the query and positive sample with $k$ hard negatives, totaling $(k + 2)N$ features. Consequently, the training cost is $2 \times (k + 2)N = (2k + 4)N$.

## E.3    OVERALL COMPLEXITY

Aggregating the costs from both stages, the total computational complexity is $(n + 6)N$ for the traditional method and $(3k + 7)N$ for the improved method. Given that the batch size $n$ is typically much larger than the number of hard negatives $k$ (i.e., $n \gg k$), our improved method significantly reduces the computational burden, particularly during the Rerank phase.

# F    THE USE OF LLMs

We only used GPT for language polishing after completing the first draft, and there were no other scenarios involving the use of Large Language Models (LLMs) in this work.

Table 14: Comparison of Computational Cost Between Traditional and Improved Distillation. $n$ represents the number of queries in a batch, and $N$ denotes the total number of queries in the dataset.

| | Item | Traditional Method | Improved Method |
|---|---|---|---|
| **Teacher model inference** | Recall | $2N$ | $2N$ |
| | Rerank | $\frac{N}{n}(n^2) = nN$ | $\frac{N}{n}(n(k+1)) = (k+1)N$ |
| | Total time(Inference Stage) | $(n+2)N$ | $(k+3)N$ |
| **Student model training** | Extract Features | $2N$ | $(k+2)N$ |
| | Total time(Training Stage) | $4N$ | $(2k+4)N$ |
| **Overall** | Overall Total | $(n+6)N$ | $(3k+7)N$ |

Table 15: U-MARVEL: Retrieval performance of each stage on M-BEIR benchmark

| Task | Dataset | Local | | | Global | | |
|---|---|---|---|---|---|---|---|
| | | Progressive transition | Hard-neg mining | Distillation | Progressive transition | Hard-neg mining | Distillation |
| $q^t \to c^i$ | VisualNews | 46.4 | 47.5 | 47.3 | 46.3 | 47.5 | 47.2 |
| | MSCOCO | 84.0 | 83.6 | 84.8 | 77.3 | 79.4 | 72.8 |
| | Fashion200K | 34.4 | 35.6 | 33.6 | 34.3 | 35.4 | 33.3 |
| $q^t \to c^t$ | WebQA | 92.4 | 95.2 | 97.1 | 92.2 | 95.0 | 96.7 |
| $q^t \to (c^i, c^t)$ | EDIS | 65.6 | 72.9 | 78.8 | 65.4 | 72.9 | 78.7 |
| | WebQA | 84.6 | 86.9 | 88.5 | 83.8 | 86.6 | 87.7 |
| $q^i \to c^t$ | VisualNews | 45.7 | 47.9 | 47.3 | 45.4 | 47.8 | 47.2 |
| | MSCOCO | 93.3 | 93.1 | 93.5 | 93.2 | 93.1 | 93.5 |
| | Fashion200K | 35.4 | 35.6 | 35.1 | 35.3 | 35.6 | 34.9 |
| $q^i \to c^i$ | NIGHTS | 31.4 | 33.0 | 34.2 | 31.4 | 33.0 | 34.0 |
| $(q^i, q^t) \to c^t$ | OVEN | 56.5 | 58.9 | 62.5 | 51.3 | 52.2 | 58.3 |
| | InfoSeek | 55.1 | 52.2 | 58.3 | 50.8 | 47.5 | 52.2 |
| $(q^i, q^t) \to c^i$ | FashionIQ | 35.4 | 34.8 | 36.4 | 35.3 | 34.7 | 36.0 |
| | CIRR | 58.7 | 57.2 | 60.7 | 55.9 | 55.2 | 56.0 |
| $(q^i, q^t) \to (c^i, c^t)$ | OVEN | 78.0 | 80.8 | 79.4 | 71.9 | 73.8 | 73.1 |
| | InfoSeek | 72.7 | 72.3 | 74.7 | 68.5 | 68.7 | 69.2 |
| | Avg. | 60.6 | 61.7 | **63.2** | 58.7 | 59.9 | **60.7** |

Table 16: Ablation study on the backbone architecture. Retrieval performance of U-MARVEL on the M-BEIR benchmark using Qwen3-VL-4B.

| Task | Dataset | Local | | | Global | | |
|---|---|---|---|---|---|---|---|
| | | Progressive transition | Hard-neg mining | Distillation | Progressive transition | Hard-neg mining | Distillation |
| $q^t \to c^i$ | VisualNews | 35.8 | 36.6 | 36.2 | 35.7 | 36.5 | 36.0 |
| | MSCOCO | 83.8 | 80.1 | 82.6 | 77.4 | 74.4 | 73.2 |
| | Fashion200K | 26.1 | 27.6 | 27.9 | 26.1 | 27.6 | 27.9 |
| $q^t \to c^t$ | WebQA | 92.9 | 95.8 | 96.8 | 92.7 | 95.6 | 96.6 |
| $q^t \to (c^i, c^t)$ | EDIS | 60.1 | 62.4 | 66.2 | 59.7 | 62.3 | 65.9 |
| | WebQA | 83.4 | 86.1 | 86.7 | 83.0 | 86.0 | 86.3 |
| $q^i \to c^t$ | VisualNews | 35.2 | 35.0 | 35.7 | 34.8 | 34.9 | 35.6 |
| | MSCOCO | 93.5 | 93.9 | 93.9 | 93.4 | 93.9 | 93.9 |
| | Fashion200K | 27.3 | 28.2 | 26.5 | 27.3 | 28.2 | 26.3 |
| $q^i \to c^i$ | NIGHTS | 32.4 | 31.9 | 33.3 | 32.3 | 31.9 | 33.2 |
| $(q^i, q^t) \to c^t$ | OVEN | 51.8 | 55.0 | 57.9 | 46.7 | 47.7 | 53.7 |
| | InfoSeek | 46.5 | 46.2 | 56.8 | 41.4 | 40.9 | 47.7 |
| $(q^i, q^t) \to c^i$ | FashionIQ | 35.4 | 34.9 | 36.0 | 35.3 | 34.8 | 35.8 |
| | CIRR | 60.3 | 54.2 | 60.5 | 57.7 | 52.8 | 57.2 |
| $(q^i, q^t) \to (c^i, c^t)$ | OVEN | 71.1 | 75.2 | 73.3 | 65.1 | 67.3 | 67.1 |
| | InfoSeek | 63.7 | 64.0 | 70.1 | 59.3 | 58.9 | 62.7 |
| | Avg. | 56.2 | 56.7 | **58.8** | 54.2 | 54.6 | **56.2** |

Table 17: Comparisons with SoTA approaches on M-BEIR benchmark in global pool setting

| Methods | $q^t \to c^i$ | | | $q^t \to c^t$ | | $q^t \to (c^i, c^t)$ | | $q^i \to c^t$ | | | $q^i \to c^i$ | | $(q^i, q^t) \to c^i$ | | $(q^i, q^t) \to c^t$ | | Avg. |
|---|---|---|---|---|---|---|---|---|---|---|---|---|---|---|---|---|---|
| | VN R@5 | CO R@5 | F200 R@10 | WQ R@5 | ES R@5 | WQ R@5 | VN R@5 | CO R@5 | F200 R@10 | NS R@5 | ON R@5 | InS R@5 | FQ R@10 | CR R@5 | ON R@5 | InS R@5 | |
| *Single model* | | | | | | | | | | | | | | | | | |
| UniIR-BLIP (Wei et al., 2024) | 23 | 75.6 | 25.4 | 79.5 | 50.3 | 79.7 | 21.1 | 88.8 | 27.6 | 33 | 38.7 | 19.7 | 28.5 | 51.4 | 57.8 | 27.7 | 45.5 |
| UniIR-CLIP (Wei et al., 2024) | 42.6 | **77.9** | 17.8 | 84.7 | 59.4 | 78.8 | 42.8 | 92.3 | 17.9 | 32 | 39.2 | 24 | 24.3 | 43.9 | 60.2 | 44.6 | 48.9 |
| MM-Embed (Lin et al., 2024) | 41 | 71.3 | 17.1 | 95.9 | 68.8 | 85 | 41.3 | 90.1 | 18.4 | 32.4 | 42.1 | 42.3 | 25.7 | 50 | 64.1 | 57.7 | 52.7 |
| LamRA-Ret (Liu et al., 2024b) | 41.3 | 75.4 | 28.7 | 85.8 | 62.5 | 81 | 39.3 | 90.4 | 30.4 | 32.1 | 48.4 | 48.7 | 33.1 | 50.5 | 70 | 60 | 54.9 |
| U-MARVEL | **47.2** | 72.8 | **33.3** | **96.7** | **78.7** | **87.7** | **47.2** | **93.5** | **34.9** | **34.0** | **58.3** | **52.2** | **36.0** | **56.0** | **73.1** | **69.2** | **60.7** |
| *+Reranker* | | | | | | | | | | | | | | | | | |
| LamRA (Liu et al., 2024b) | 46.9 | **78** | 32.5 | 96.5 | 74.4 | 87.1 | 47.6 | 92.4 | 36.6 | 34.2 | 54 | **58.7** | 37.4 | **59.7** | 72.6 | **74** | 61.4 |
| U-MARVEL⁺ | **48.8** | 70.1 | **33.8** | **98.3** | **80.8** | **88.3** | **49.8** | 86.0 | **36.8** | **34.8** | **58.7** | 56.9 | **37.4** | 58.4 | **74.9** | 73.8 | **61.8** |

Table 18: Held-out task generalization experiments were conducted on M-BEIR. The asterisk * indicates that the model was trained on the remaining five tasks, without any exposure to the three held-out tasks.

| Methods | $q^i \to c^i$ | $(q^i, q^t) \to c^t$ | | $(q^i, q^t) \to (c^i, c^t)$ | | Local Avg. |
|---|---|---|---|---|---|---|
| | NIGHTS R@5 | OVEN R@5 | InfoSeek R@5 | OVEN R@5 | InfoSeek R@5 | |
| *Supervised* | | | | | | |
| UniIR-BLIP (Wei et al., 2024) | 33.0 | 41.0 | 22.4 | 55.8 | 33.0 | 37.0 |
| UniIR-CLIP (Wei et al., 2024) | 32.0 | 45.5 | 27.9 | 67.6 | 48.9 | 44.4 |
| *Zero-shot* | | | | | | |
| LamRA-Ret* (Liu et al., 2024b) | 27.2 | **44.7** | 44.0 | **62.8** | 49.5 | 45.6 |
| U-MARVEL* | **31.5** | 44.3 | **49.2** | 62.6 | **52.6** | **48.0** |
| LamRA* (Liu et al., 2024b) | 29.2 | 46.9 | 54.2 | **65.1** | 59.1 | 50.9 |
| U-MARVEL⁺* | **32.1** | **49.0** | **55.3** | 64.9 | **61.2** | **52.5** |

