# OpenReview forum: "U-MARVEL: Unveiling Key Factors for Universal Multimodal Retrieval via Embedding Learning with MLLMs"
_ICLR.cc/2026/Conference — ICLR 2026 Poster_

### Official Review · Reviewer_SuPC · 2025-10-22

**Soundness:** 2
**Presentation:** 2
**Contribution:** 2
**Rating:** 4
**Confidence:** 5

**Summary:**

This paper investigates key design decisions for universal multimodal retrieval (UMR) with MLLMs.
The analysis spans three axes: (1) embedding generation choices, (2) Contrastive learning training factors (i.e., batch size, learning rate, temperature; hard-negative mining), and (3) re-ranker distillation into a single model.
Based on these findings, the authors propose U-MARVEL, a unified embedding-learning framework that targets single-model retrieval on M-BEIR.

**Strengths:**

1. Practically impactful problem: design decisions in universal multimodal retrieval are often overlooked and this work advocates for more attention on this, in which can be a good initiative for UMR deployability.
2. Performance: the proposed U-MARVEL framework improves retriever performance on M-BEIR, particularly in single model retrieval.

**Weaknesses:**

1. Writing clarity: some paragraphs can be revised to improve readability. (e.g., L44: define concrete examples for “diverse and complex requirements in the real-world” in the intro; L161: state the exact metric used for Table 1 (and other tables))
2. Evidence strength: e.g., Gains of 0.1/0.3 (ID-0 vs ID-1) are tiny -- consider significance tests or a larger sample. L263: The square-root rule is argued via only two datapoints (ID-0 vs ID-2); The “from simple to complex” motivation would benefit from a curriculum-learning citation or an ablation isolating each step’s contribution.
3. Some motivations needs to be justified: e.g., L224: "following the principle of advancing from simpler to more complex tasks" would benefit from either a curriculum-learning citation or an ablation isolating each step’s contribution.

**Questions:**

1. L413, 414: there should be a space between method name and inline citation.
2. L268-269: please make the claim ("we argue that...") more rigorous or use evidence to back it.
3. How exactly do you filter hard negatives? (i.e., what threshold value and how do you select the threshold)
4. Can you provide empirical results (e.g., latency) to compare efficiency between traditional distillation and the improved distillation?
5. Please fix the typo "exihibits".

---

> ### Author Response · Authors · 2025-11-26
>
> > Response to W1:
>
> We thank the reviewer for the careful reading and specific suggestions to improve the manuscript's readability. We have revised the paper accordingly:
>
> 1. **Clarifying "Complex Requirements" (Introduction):** We agree that providing concrete examples helps readers better understand the challenges in real-world scenarios. We have revised the sentence in the Introduction to explicitly specify these requirements using the following phrasing:
>
>    - **Revised Text:** "...they often struggle to address the diverse and complex requirements encountered in real-world scenarios, ranging from fine-grained instruction following to multi-turn interleaved interactions."
>
> 2.  **Clarifying Metrics (Table 1 & others):** We apologize for not explicitly stating the metric definition in the main text near the tables. We have added a global definition statement at the beginning of Section 3. We explicitly state that all ablation studies (Tables 1–6) report the Average Recall, calculated as the mean of R@5 and R@10 across the benchmark tasks. This ensures clarity for Table 1 and all subsequent empirical results without redundancy.
>
> 3. **General Polish:** We have also conducted a thorough proofreading to correct typos (e.g., "exhibits") and fix spacing issues between method names and citations throughout the manuscript.
>
> ---
>
> > Response to W2:
>
> We thank the reviewer for scrutinizing the strength of our empirical evidence. We address these three specific concerns below:
>
> 1. **Significance of Small Gains in Instruction Masking (Table 2):**
>
>    We acknowledge that the performance gain of masking instruction tokens (0.1% Local / 0.3% Global) appears marginal numerically. The primary motivation for "Masking Instructions" is theoretical soundness rather than raw metrics. As hypothesized in Section 3.1.2, unmasked instruction tokens introduce calculation bias into the pooled embedding. By removing them, we ensure the embedding represents the semantic content of the input rather than the instruction pattern. In addition, the fact that masking removes potential bias without degrading performance (and even slightly improving it) validates it as a safer, more robust design choice compared to including them.
>
> 2. **Evidence for Learning Rate Scaling (Table 4):**
>
>    We agree that deriving a scaling law from two data points is statistically limited. However, our claim is not that we derived the square-root rule, but that our experiments verify its applicability to MLLM embedding training. As shown in Table 4, simply quadrupling the batch size without LR scaling (ID-1) yielded negligible gains (57.2% --> 57.4%). In contrast, applying the scaling rule (ID-2) significantly boosted performance to 58.3%. This comparison (ID-1 vs. ID-2) provides strong evidence that learning rate scaling is essential. We have revised the text to clarify that this observation aligns with established scaling laws in deep learning (Goyal et al., 2017) rather than being a novel derivation from limited data.
>
> ---
>
> > Response to W3:
>
> We thank the reviewer for highlighting the need for stronger justification for our Progressive Transition strategy. We agree that a key methodological motivation like "advancing from simpler to more complex tasks" requires robust theoretical and empirical evidence. We have addressed this through two main actions, satisfying both requests:
>
> 1. **Theoretical Justification:**
>
>    We have added a citation to the foundational work on Curriculum Learning (Bengio et al., 2009) in Section 3.1.3. This theoretically grounds our motivation, explaining that this strategy is designed to guide the optimization process toward better local minima by starting with simpler semantic alignment tasks before moving to complex cross-modal instructions.
>
> 2. **Ablation Clarification:**
>
>    We respectfully point out that Table 3 is explicitly designed as the ablation study the reviewer requested. It isolates each step's contribution:
>
>    - **ID-0:** Baseline (Instruction Tuning only).
>
>    - **ID-1:** Isolates the contribution of Text Adaptation (+0.7% Local / +1.6% Global).
>
>    - **ID-2:** Isolates the contribution of Cross-modal Alignment (+0.4% Local / +0.3% Global).
>
>    This stepwise breakdown quantifies exactly how the progressive transition builds up performance.

---

> > ### Author Response · Authors · 2025-11-26
> >
> > > Response to Q1:
> >
> > We thank the reviewer for attention to detail. We have carefully proofread the manuscript and corrected the spacing issues between method names and inline citations in Table 7.
> >
> > ---
> >
> > > Response to Q2:
> >
> > We appreciate the reviewer's suggestion to strengthen this claim. The statement "we argue that the learnable temperature parameter... dynamically adjusts the sharpness... enabling the model to optimize this critical hyperparameter" is supported by our empirical findings in Section 3.2.1. As shown in Table 4, we conducted a direct comparison between fixed temperature and learnable temperature settings. Specifically, comparing ID-4/ID-6 (fixed temperature) with ID-7/ID-8 (learnable temperature) reveals a consistent performance improvement (e.g., increasing from 58.9% to 60.3% in Local Avg). We have revised the text in Sec 3.2.1 from the manuscript.
> >
> > ---
> >
> > > Response to Q3:
> >
> > 1. **Hard-Negative Filtering Details and Rationale**
> >
> >    In our experiments, we set k=5 and the predefined threshold to 0.7; specifically, any negative sample with a score exceeding 0.7 is filtered out. The selection of k=5 for the hard negative mining stage represents a trade-off between training efficiency and sample quality. The threshold was selected based on the cosine similarity distribution of the embedding space observed after the "Progressive Transition" stage.
> >
> >    We found that true positive pairs typically exhibit similarity scores significantly greater than 0.7. Conversely, negative samples that also score above 0.7 are highly indicative of false negatives—semantically identical items incorrectly labeled as negatives due to inherent dataset noise. Treating these as negatives would force the model to separate semantically equivalent concepts, which severely compromises convergence (as detailed in Finding 5). Therefore, threshold=0.7 serves as a critical cutoff to effectively filter out label noise while preserving the "hardest valid negatives" (which typically fall within the 0.3 - 0.7 range).
> >
> > 2. **On Dataset Specificity**
> >
> >    The threshold is not dataset-specific; it is designed to be a universal parameter. While we acknowledge that fine-tuning threshold for each individual dataset might yield marginally higher metrics, we prioritize a unified setting to align with U-MARVEL's goal of universal retrieval. Our consistent performance across the M-BEIR benchmark demonstrates that a fixed threshold=0.7 remains highly effective, proving the model's ability to adapt robustly to diverse data distributions rather than relying on brittle, dataset-specific tuning.
> >
> > 3. **Sensitivity Analysis:**
> >
> >    The model exhibits strong robustness around the threshold of 0.7 (specifically within the range [0.6, 0.75]). Setting the threshold too high (>0.85) fails to filter false negatives, causing convergence issues and performance degradation similar to the unfiltered setting observed in Table 5 (ID-1). Conversely, setting it too low (<0.5) discards informative hard negatives, reducing the task to trivial discrimination and diminishing learning efficiency.
> >
> > ---
> >
> > > Response to Q4:
> >
> > We clarify that conducting a direct empirical latency measurement for the "Traditional Distillation" was not feasible because, as noted in Section 3.3, the computational cost of the traditional approach is prohibitive for large-scale MLLM training settings ($O(Nn)$ complexity). However, we have provided a rigorous quantitative comparison in Table 14 (Appendix E) using the actual hyperparameters from our experiments.
> >
> > - **Setup:** Based on our training configuration (Batch size n = 3840, Hard negatives k = 50).
> >
> > - **Result:** The theoretical calculation demonstrates that our improved distillation method reduces the computational load to approximately 4.1% of the traditional method.
> >
> > - **Interpretation:** This corresponds to a ~25x speedup, transforming a computationally impractical task into a feasible one. Since the operation time is dominated by the number of forward/backward passes (which are linear to the complexity terms analyzed), this calculated ratio serves as an accurate proxy for empirical latency improvements.
> >
> > ---
> >
> > > Response to Q5:
> >
> > We apologize for this oversight. We have corrected the typo "exihibits" to "exhibits" in the Abstract and have conducted a thorough spell-check across the entire manuscript.

---

### Official Review · Reviewer_WToA · 2025-10-29

**Soundness:** 2
**Presentation:** 2
**Contribution:** 1
**Rating:** 4
**Confidence:** 5

**Summary:**

This paper presents a systematic empirical study on using multimodal large language models as embedding models for Universal Multimodal Retrieval. The authors implement a baseline contrastive learning pipeline (Qwen2-VL-7B with LoRA) and investigate three main axes: (1) adapting decoder-only MLLMs into instruction-aware embedders (embedding extraction strategies, instruction handling, progressive transition); (2) training MLLM-based embedders under InfoNCE (batch size / learning rate / temperature interactions, hard negative mining and filtering); and (3) compressing the recall-then-rerank pipeline into a single model via improved distillation. Key findings include that bidirectional attention + mean pooling outperforms last-token/compression prompt schemes, masking instruction tokens during pooling helps, learnable temperature improves contrastive learning, and filtered hard negatives stabilize training. Based on these insights the authors propose U-MARVEL, and demonstrate substantial gains on M-BEIR and many zero-shot benchmarks.

**Strengths:**

1.	the paper offers actionable, well-motivated findings (embedding extraction, instruction masking, progressive transition, filtering of hard negatives, distillation).
2.	backbone, datasets, and training configs are explicitly listed (Tables in Appendix). This makes replication feasible.

**Weaknesses:**

1.	Limited novelty, this work is just a conbination of existing tricks in retreiver training, some techs are already presented and disscussed by previous works, such as mean token and bidirection attention [1], learnable temperature [2]. Porting them from LLM to MLLM offers little scientific significance.
2.	Insufficient detail on hard-negative filtering criteria, the choice and sensitivity of the threshold for filtering presumed false negatives needs more analysis.
3.	Some experimental comparisons are not entirely fair, and certain conclusions appear to be inconsistent across experiments. Please refer to the Questions section for detailed discussion.
4.  Lacks sufficient details regarding experimental implementation, parameter selection, and description of observed phenomena, such as why masking instruction tokens during mean pooling enhances embedding performance. In such cases, a deeper analysis is required; merely stating the results is insufficient for a valid research contribution.

[1] LLM2Vec: Large Language Models Are Secretly Powerful Text Encoders (BehnamGhader, COLM 2024)

[2] Analyzing the Impact of Learnable Softmax Temperature in Contrastive Visual-Textual Alignment Systems: Benefits, Drawbacks, and Alternative Approaches (Sun and Li, TMLR 2024)

**Questions:**

1.	In Section 3.1.2, the paper discusses the effectiveness of multi-stage training. However, it is not clear whether this effectiveness comes from the increased amount of training data or from the multi-stage training strategy itself. Is the comparison and the resulting conclusion objective and fair under this ambiguity?
2.	In Section 3.1.1, the paper states that the presence or absence of the Compression Prompt has a significant impact on model performance. Is this effect truly as substantial as reported? Based on my experience, the impact may not be that pronounced.
3.	How exactly is the similarity threshold chosen for filtering false negatives in hard-negative mining? Is it dataset-specific, and how sensitive are results to this threshold?
4.	What is the initialization and optimizer settings of the learnable temperature? Do you observe stable convergence of τ across different batch sizes? Has the variation process of the temperature been recorded？
5.	In the recall-then-rerank pipeline, what is the value of α? Does it vary during training or remain fixed? How is its magnitude determined?
6.	In Section 3.2.2, what is the value of k used in the top-k selection? There are several unclear parameter choices throughout the paper — could the authors specify the exact values of these parameters and the criteria used for their selection?
7.	In Section 3.3, the “continue-hard” experiments show that using only hard negatives leads to performance improvement, while in Section 3.2.2, using only hard negatives resulted in a failure. These conclusions appear inconsistent — is this discrepancy caused by differences in experimental settings or by the choice of data?
8.	The distillation method proposed in Section 3.3 appears relatively conventional — can it truly be considered an innovation? Moreover, the paper only provides a theoretical derivation of the time savings. How does the proposed method’s performance compare with traditional approaches in practice, and is there any empirical comparison of the actual time consumption?

---

> ### Author Response · Authors · 2025-11-26
>
> > Response to W1:
>
> We thank the reviewer for pointing this out. However, we respectfully disagree with the assessment that this work offers "limited novelty" or is merely a "combination of existing tricks." While we acknowledge that individual concepts like mean pooling or learnable temperature exist in broader deep learning literature, our contribution lies in the systematic discovery of why and how these components must be adapted to bridge the gap between generative MLLMs and discriminative retrieval tasks.
> We clarify the scientific significance and methodological novelty of our contributions as follows:
>
> 1. **Novelty in Discovery: Correcting Suboptimal Design Principles.**
>
>    Scientific novelty lies not only in proposing new architectures but also in rigorously identifying why certain designs succeed where others fail.
>
>    - **Correcting the "Last Token" Bias**: Current SoTA methods (e.g., LamRA, VLM2Vec) predominantly rely on "Last Token" pooling with compression prompts. Our work provides the first systematic empirical evidence in the MLLM-UMR context that this standard practice is suboptimal. We demonstrate that Bidirectional Attention with Mean Pooling is the superior paradigm. This is not a trivial "port"; it is a correction of a prevailing architectural flaw in the community.
>
>    - **Instruction Masking**: We introduce a specific "Masked Instruction" mechanism during pooling. This is a targeted architectural modification to mitigate the bias introduced by instruction tokens during feature aggregation, which resulted in observable performance gains.
>
> 2. **Novelty in Training Recipes: Addressing MLLM-Specific Challenges.**
>
>    Merely "porting" tricks from LLMs often leads to failure in multimodal contexts. We introduce specific methodological solutions to address these unique adaptation challenges:
>
>    - **Progressive Transition Strategy**: We propose a non-standard, three-stage curriculum (Text --> Image-Text --> Multimodal Instruction). This is designed specifically to counteract the catastrophic forgetting and capability imbalance that arise when adapting a large generative MLLM to a specialized discriminative retrieval task.
>
>    - **False Negative Filtering in Hard Negative Mining**: We found that standard hard negative mining techniques led to "complete failure" (model collapse) in our context. Consequently, we designed a specific threshold-based filtering mechanism mixed with in-batch negatives, turning a failed training run into a state-of-the-art result.
>
>    - **Interaction of Hyperparameters**: Regarding learnable temperature, our study reveals its critical dependency on large batch sizes and learning rate scaling. We verify that in the specific context of MLLM contrastive learning, this parameter is not just a "trick" but a necessity for stability.
>
> 3. **Novelty in Efficiency: Improved Knowledge Distillation**
>
>    As the reviewer kindly acknowledged in other comments, our distillation method is a technical innovation. We transform the computationally prohibitive teacher-student alignment process—typically $\mathcal{O}(Nn)$—into a feasible $\mathcal{O}(Nk)$ operation. This yields a calculated ~25x speedup (reducing total time to 4.1% of the original), making the distillation of sophisticated rerankers into single-stage retrievers practically feasible for the first time in this domain.
> In summary, U-MARVEL is not a random assembly of existing components. It is a unified framework built on systematic experimental discoveries (Finding 1-6) that corrects suboptimal practices and introduces novel training strategies (Progressive Transition, Improved Distillation) to set a new state-of-the-art for universal multimodal retrieval.

---

> > ### Author Response · Authors · 2025-11-26
> >
> > > Response to W2:
> >
> > Please refer to the response of Q3
> >
> > ---
> >
> > > Response to W3:
> >
> > Please refer to the responses of Q1.
> >
> > ---
> >
> > > Response to W4:
> >
> > We sincerely appreciate the reviewer’s constructive feedback regarding the depth of our experimental details and the analysis of observed phenomena. We agree that in the initial version, the explanation of why certain methods work (such as masking instruction tokens) was not sufficiently explored, and that key implementation details were not prominent enough.
> >
> > We have extensively revised the manuscript to improve self-containment and analytical depth, with all major changes marked in red.
> >
> > **First,** to address concerns regarding method clarity, we relocated core mathematical formulations and implementation details from the appendix to the main text (Section 3). This includes the formal definitions of the InfoNCE and Distillation losses, as well as critical hyperparameters such as the false negative filtering thresholds and the recall-rerank fusion weighting parameter alpha.
> >
> > **Second,** we have integrated empirical results previously isolated in the appendices into the main discussion to provide a more holistic evaluation. Specifically, we incorporated the Ablation Study and Global Pool settings into Section 4 to highlight the specific contributions of each training stage and the model’s robustness in large-scale retrieval. We also strengthened our claims by adding new experiments in Appendix D across different backbones to verify generalizability and included a dedicated description of our reproducibility protocols regarding experimental randomness in Appendix C.
> >
> > **Third,** we rewrote the analysis in Sections 4.1 and 4.2 to move beyond mere numerical reporting toward mechanistic interpretation. We now offer deep-dive analyses on the necessity of learnable temperature coefficients in large-batch training and the evolution of model capabilities across different data sources. Furthermore, we explicitly discuss the trade-off between efficiency and effectiveness compared to cascade systems.

---

> > > ### Author Response · Authors · 2025-11-26
> > >
> > > > Response to Q1:
> > >
> > > We thank the reviewer for raising this incisive question regarding the rigorous attribution of our Progressive Transition's effectiveness. We agree that distinguishing between the impact of data quantity and strategic sequencing is essential for a fair conclusion. We clarify that the comparison is indeed objective and fair, as the benefit is demonstrably derived from the training strategy and its specific sequence.
> > >
> > > 1. **The strategy's initial stage (ID-1) already surpasses SOTA**:
> > >
> > >    The most compelling evidence lies in comparing our first stage (ID-1) against existing single-model baselines:
> > >
> > >    - **Our ID-1 Strategy**: The model trained only with the initial Text Adaptation stage achieves a Local Avg of 57.3% and a Global Avg of 55.5% (Table 3).
> > >
> > >    - **Previous Single-Model SOTA (LamRA-Ret)**: The strongest comparable single-model baseline achieves a Local Avg of 56.6% (Table 7) and a Global Avg of 54.9% (Table 17).
> > >
> > >    This comparison is highly instructive: ID-1, which represents only the first strategic step of our Progressive Transition, already surpasses the state-of-the-art single-model performance using the same backbone and training data. This definitively proves that the strategic sequencing—i.e., adapting the model with text-only data first—is effective and yields significant gains immediately, irrespective of the full data volume used in later stages.
> > >
> > > 2. **The full gain is attributed to the curriculum learning**:
> > >
> > >    As shown in Table 3, the observed cumulative gain of 1.9% (Global Avg: ID-0 --> ID-2) is therefore a result of the curriculum learning strategy—sequencing tasks from simpler (Text Adaptation) to more complex (Cross-modal Alignment and Full Instruction). The Progressive Transition design is a holistic recipe that ensures the effective adaptation of decoder-only MLLMs to embedding tasks.
> > >
> > > ---
> > >
> > > > Response to Q2:
> > >
> > > We thank the reviewer for this insightful observation. We largely agree with your assessment that the Compression Prompt itself is not a universal performance booster. We would like to clarify that our finding regarding its "significant impact" highlights its architectural dependency rather than its general superiority. Our experiments in Table 1 and Section 3.1.1 reveal two key nuances that validate your intuition:
> > >
> > > 1. **The prompt is only critical for "Last Token" architectures (ID-0 vs. ID-2)**: We found that the Compression Prompt is strictly coupled with the Causal Attention and Last Token mechanism. As shown in Table 1, the standard approach (ID-0) achieves a Local Avg of 56.6. However, if this prompt is used with a mismatched architecture (e.g., Mean Token), the performance drops drastically to 33.7 (ID-2). This confirms that the prompt is a necessary "patch" to force summarization in autoregressive models, rather than a generally effective feature.
> > >
> > > 2. **A simpler architecture without the prompt works better (ID-0 vs. ID-4)**: Consistent with your experience, our results show that the prompt becomes redundant when the architecture is optimized. By simply switching to Bidirectional Attention with Mean Pooling and removing the Compression Prompt entirely (ID-4), we achieved a Local Avg of 57.2. This represents a modest improvement over the complex prompt-based baseline (ID-0, 56.6).
> > >
> > > In summary: We clarify that the "significant impact" refers to the prompt's role in rescuing the performance of legacy "Last Token" mechanisms. However, our proposed U-MARVEL framework demonstrates that removing the prompt in favor of a simpler Bidirectional + Mean Pooling strategy yields superior results.

---

> > > > ### Author Response · Authors · 2025-11-26
> > > >
> > > > > Response to Q3:
> > > >
> > > > 1. **Hard-Negative Filtering Details and Rationale**:
> > > >
> > > >    We set the similarity threshold to 0.7. The threshold was selected based on the cosine similarity distribution of the embedding space observed after the "Progressive Transition" stage. We found that true positive pairs typically exhibit similarity scores significantly greater than 0.7. Conversely, negative samples that also score above 0.7 are highly indicative of false negatives—semantically identical items incorrectly labeled as negatives due to inherent dataset noise. Treating these as negatives would force the model to separate semantically equivalent concepts, which severely compromises convergence (as detailed in Finding 5). Therefore, threshold=0.7 serves as a critical cutoff to effectively filter out label noise while preserving the "hardest valid negatives" (which typically fall within the 0.3 - 0.7 range).
> > > >
> > > > 2. **On Dataset Specificity**
> > > >
> > > >    The threshold is not dataset-specific; it is designed to be a universal parameter. While we acknowledge that fine-tuning threshold for each individual dataset might yield marginally higher metrics, we prioritize a unified setting to align with U-MARVEL's goal of universal retrieval. Our consistent performance across the M-BEIR benchmark demonstrates that a fixed threshold=0.7 remains highly effective, proving the model's ability to adapt robustly to diverse data distributions rather than relying on brittle, dataset-specific tuning.
> > > >
> > > > 3. **Sensitivity Analysis**:
> > > >
> > > >    The model exhibits strong robustness around the threshold of 0.7 (specifically within the range [0.6, 0.75]). Setting the threshold too high (>0.85) fails to filter false negatives, causing convergence issues and performance degradation similar to the unfiltered setting observed in Table 5 (ID-1). Conversely, setting the threshold too low (<0.5) discards informative hard negatives, reducing the task to trivial discrimination and diminishing learning efficiency.

---

> > > > > ### Author Response · Authors · 2025-11-26
> > > > >
> > > > > > Response to Q4:
> > > > >
> > > > > We thank the reviewer for this insightful question regarding the implementation details of the learnable temperature parameter. We are happy to provide the specific settings and our observations regarding its convergence behavior. We have added these details and the analysis results into Section 3.2.1 and Appendix D in the revised manuscript.
> > > > >
> > > > > 1. **Initialization and Optimizer Settings**
> > > > >
> > > > >    As detailed in Table 12 from Appendix C, we initialize the learnable temperature at 0.05 at the beginning of the "Progressive Transition (stage-3)" (where we switch from fixed to learnable temperature). The temperature scalar is optimized jointly with the model parameters using the AdamW optimizer. We treat $\tau$ as a learnable parameter within the loss function (Eq. 1), allowing it to dynamically adjust the scale of the logits during the training process.
> > > > >
> > > > > 2. **Stability Across Batch Sizes**
> > > > >
> > > > >    We do observe stable convergence of $\tau$ across different settings.
> > > > >
> > > > >    - **Stable Convergence Across Batch Sizes**: In Table 4, for ID-7 (batch size 3840), the temperature parameter ($\tau$) converges to 0.0206 (as further detailed in Table 12 under "Progressive Transition (stage-3)"), while for ID-8 (batch size 7680), it converges to 0.0195. This indicates that the temperature slightly decreases as the batch size increases, which aligns with the findings cited in the referenced literature [2]: larger batch sizes introduce more noise, necessitating a sharper probability distribution to effectively handle the increased complexity and ambiguity in the training data.
> > > > >
> > > > >    - **Consistent Performance Enhancement**: As shown in Table 4, a learnable temperature parameter yielded significant and consistent performance gains in both large-batch settings: ID-7 (batch size 3840) and ID-8 (batch size 7680) achieved local averages of 60.1 and 60.3, respectively. These results surpass those of their fixed-temperature counterparts (ID-4 and ID-6, both at 58.9). The consistent improvement demonstrates the robustness of the learnable temperature mechanism, which converges effectively across different batch conditions by dynamically adapting to the optimization landscape.
> > > > >
> > > > >    - **Superiority Over Manually Tuned Fixed $\tau$**: To further validate the effectiveness of the learnable temperature parameter, we conducted additional experiments where we fixed $\tau$ at 0.02—aligning closely with the final converged value of 0.0206 observed in our learnable $\tau$ setting (Table 12, "Progressive Transition (stage-3)"). The results demonstrated that the fixed $\tau$ configuration achieved a performance metric of 59.8, whereas the learnable $\tau$ achieved a higher metric of 60.1 (Table 4, ID-7). This indicates that the learnable $\tau$ not only exhibits greater robustness to different initializations but also outperforms even a carefully selected fixed $\tau$ value.
> > > > >
> > > > > 3. **Variation Process and Recording**
> > > > >
> > > > >    Yes, we have recorded the variation of the temperature throughout the training stages. As reported in the "Temp ($\tau$)" column of Table 12, the temperature consistently converges to a value near 0.02, specifically:
> > > > >
> > > > >    - Stage-3 (Instruction-tuned Multimodal Retrieval): Converges from 0.05 --> 0.0206.
> > > > >
> > > > >    - Hard-negative Mining: Converges from 0.0206 --> 0.0188.
> > > > >
> > > > >    - Distillation: Converges from 0.0188 --> 0.0204.
> > > > >
> > > > >    Under the Hard-negative Mining strategy, the temperature parameter ($\tau$) experiences a further decrease. A lower $\tau$ makes the loss function more sensitive to the gradients from hard negative samples.

---

> > > > > > ### Author Response · Authors · 2025-11-26
> > > > > >
> > > > > > > Response to Q5:
> > > > > >
> > > > > > Thank you for pointing this out. In our recall-then-rerank pipeline, the value of $\alpha$ is set to 0.5. This parameter remains fixed and does not vary during training. Regarding the determination of its magnitude, we follow the configuration adopted in LamRA (Liu et al., 2024b). By setting $\alpha = 0.5$, we assign the 1:1 weight ratio to the recall scores and rerank scores.
> > > > > >
> > > > > > This implies that we treat the retrieval scores (cosine similarity) and the reranking scores (prediction probability) as equivalent and equally important contributions to the final ranking. We found that this balanced combination effectively leverages the strengths of both stages without requiring complex weight tuning.
> > > > > >
> > > > > > > Response to Q6:
> > > > > >
> > > > > > Thank you for pointing this out. We apologize for the ambiguity regarding the parameter settings. The value of $k$ varies depending on the specific training stage (Hard Negative Mining vs. Distillation) to serve different objectives. We clarify the exact values and the criteria for their selection below:
> > > > > >
> > > > > > 1. **In Section 3.2.2 (Hard Negative Mining)**: We set $k=5$. The selection of $k=5$ for the hard negative mining stage is determined by a trade-off between training efficiencynd sample quality
> > > > > >
> > > > > >    - **Computational Efficiency**: Training on the large-scale M-BEIR dataset (1.33M queries) is computationally intensive. On our infrastructure (64 Ascend-910B GPUs), a single epoch of standard training takes approximately 3 hours. Incorporating explicit hard negatives significantly increases the computational load because it requires processing additional candidate embeddings. We observed that setting $k=5$ increases the training time by approximately 3.5 times compared to the baseline. Increasing $k$ further would make the training cost prohibitively high without proportional performance gains.
> > > > > >
> > > > > > 2. **In Section 3.3 and Figure 2 (Reranker Distillation)**: We set $k=50$. This decision is driven by the inference pipeline and evaluation metrics:
> > > > > >
> > > > > >    - **Alignment with Inference**: During the inference phase of the recall-then-rerank pipeline, we first retrieve the top-50 candidates using the recall model and then re-rank them using the reranker. To effectively distill this entire pipeline into a single embedding model, the student model must learn from the same distribution of candidates that the reranker sees. Therefore, we use the top-50 hard negatives to construct the distillation samples, ensuring the student model learns to mimic the reranker's discrimination ability over the entire candidate pool used during inference.
> > > > > >
> > > > > >    - **Metric Coverage**: Our evaluation metrics include Recall@1, Recall@5, Recall@10 and Recall@50. Setting $k=50$ ensures that the optimization objective covers the scope of all reported metrics, allowing the model to improve ranking performance across the entire top-50 list.
> > > > > >
> > > > > > We will include these specific parameter values and their justifications in the implementation details section of the final revision.

---

> > > > > > > ### Author Response · Authors · 2025-11-26
> > > > > > >
> > > > > > > > Response to Q7:
> > > > > > >
> > > > > > > The discrepancy is primarily due to two factors: the noise level in the data distribution (determined by $k$) and the model's initialization state (curriculum learning).
> > > > > > >
> > > > > > > 1. **Data Distribution & Noise ($k = 5$ vs. $k = 50$)**:
> > > > > > >
> > > > > > >    As noted in Section 3.2.2, a major challenge in hard negative mining is the presence of false negatives, which can hinder convergence. In Section 3.2.2, the failed experiment used only the top-$k$ ($k = 5$) negatives. These samples are extremely "hard" and contain a high density of false negatives (semantic matches labeled as negatives), causing the training to collapse. In Section 3.3, the "Continue-hard" experiment utilized the distillation dataset, which consists of top-$k$ ($k=50$) negatives. This broader $k$ range introduces a "softer" distribution of hard negatives, significantly diluting the noise from false negatives and making the optimization landscape smoother.
> > > > > > >
> > > > > > > 2. **Curriculum Learning (Model Initialization)**:
> > > > > > >
> > > > > > >    The starting points for these experiments differ significantly. The failure in Section 3.2.2 occurred when applying hard negatives to a model that had only completed progressive transition. In contrast, the "Continue-hard" experiment in Section 3.3 (Table 6) is initialized with the converged Recall Model from Section 3.2.2. This model has already adapted to mixed hard negatives and possesses robust discriminative capabilities, enabling it to benefit from further training on hard negatives without collapsing.

---

> > > > > > > > ### Author Response · Authors · 2025-11-26
> > > > > > > >
> > > > > > > > > Response to Q8:
> > > > > > > >
> > > > > > > > 1. **Innovation**: Making Reranker Distillation Feasible for MLLMs
> > > > > > > >
> > > > > > > >    While the use of KL-divergence for distillation is established, our contribution lies in the efficient data construction strategy tailored for the "Recall-then-Rerank" paradigm in MLLMs.
> > > > > > > >
> > > > > > > >    - **The Challenge**: Traditional distillation typically requires the teacher (Reranker) to compute scores for all pairs in a batch (e.g., an $n \times n$ matrix). For MLLM-based Rerankers (which act as Cross-Encoders), this incurs a computational complexity of $O(n^2)$, making it computationally prohibitive for large-scale training.
> > > > > > > >
> > > > > > > >    - **Our Solution**: We propose a topology-aware distillation method. Instead of a full matrix, we reconstruct the training data into triplets of (query, positive, top-$k$ hard negatives). This aligns the distillation process with the inference logic (retrieving top candidates then reranking) and reduces the Reranker's complexity from quadratic to linear $O(n \cdot k)$. It reduces computational overhead while increasing feature diversity by select top-$k$ hard negatives.
> > > > > > > >
> > > > > > > >    - **Key Insight**: This is not merely an optimization but a necessary adaptation that enables the distillation of massive MLLM Rerankers into Retrievers—a process that was previously impractical due to resource constraints.
> > > > > > > >
> > > > > > > > 2. **Empirical Time Comparison**
> > > > > > > >
> > > > > > > >    We clarify that conducting a direct empirical latency measurement for the "Traditional Distillation" was not feasible because, as noted in Section 3.3, the computational cost of the traditional approach is prohibitive for large-scale MLLM training settings ($O(Nn)$ complexity). However, we have provided a rigorous quantitative comparison in Table 14 (Appendix E) using the actual hyperparameters from our experiments.
> > > > > > > >
> > > > > > > >    - **Setup:** Based on our training configuration (Batch size $n = 3840$, Hard negatives $k = 50$).
> > > > > > > >
> > > > > > > >    - **Result:** The theoretical calculation demonstrates that our improved distillation method reduces the computational load to approximately 4.1% of the traditional method.
> > > > > > > >
> > > > > > > >    - **Interpretation:** This corresponds to a ~25x speedup, transforming a computationally impractical task into a feasible one. Since the operation time is dominated by the number of forward/backward passes (which are linear to the complexity terms analyzed), this calculated ratio serves as an accurate proxy for empirical latency improvements.
> > > > > > > >
> > > > > > > >    In our experiments (64 GPUs, Top-50 hard negatives, 10% sample data), our method requires 5 hours for inference and 9 hours for training. Consequently, the traditional method is projected to require approximately $(5 + 9) \times 25 = 350$ hours—a duration that is excessively long for practical implementation.
> > > > > > > >
> > > > > > > > 3. **Performance Comparison**
> > > > > > > >
> > > > > > > >    The traditional method failed to complete valid training steps due to computational timeouts. We evaluate our method's effectiveness by comparing it against strong baselines and the Teacher model itself (c.f. Table 6).

---

> ### Comment · Reviewer_WToA · 2025-11-26
>
> Thanks for the detailed response from authors. It solves some of my concerns and questions. However, i hold the opinion about the novelty.
>
> ---
>
> > We introduce a specific "Masked Instruction" mechanism during pooling.
>
> This strategy is already implemented in `sentence_transformers` and code of GritLM (arxiv 2402.09906, ICLR 2025) I think.
>
> > Novelty in Training Recipes: Addressing MLLM-Specific Challenges. False Negative Filtering in Hard Negative Mining. Interaction of Hyperparameters
>
> Why do these existing problems need to be solved again in the context of MLLM? Do they actually bring any new challenges? In your work, shifting from a PLM to an LLM is basically just switching to a stronger backbone. The overall setup doesn’t really seem to change in any essential way.

---

### Official Review · Reviewer_4154 · 2025-10-30

**Soundness:** 3
**Presentation:** 3
**Contribution:** 3
**Rating:** 4
**Confidence:** 4

**Summary:**

The paper introduces a novel framework, U-MARVEL, aimed at improving Universal Multimodal Retrieval (UMR) using Multimodal Large Language Models (MLLMs). While current state-of-the-art UMR methods have made significant strides, they often face limitations in terms of retrieval capability, generalization, and the underlying mechanisms driving performance. The authors systematically explore the key factors that contribute to effective embedding learning and retrieval performance, such as embedding generation strategies, progressive training transitions, hard negative mining, and re-ranker distillation. They present U-MARVEL, a unified framework that integrates these findings, demonstrating superior performance over existing methods on the M-BEIR benchmark and strong zero-shot capabilities for tasks like text-to-video retrieval.

**Strengths:**

1. The introduction of U-MARVEL as a unified framework is a significant contribution. It successfully integrates multiple advanced techniques to create a more efficient and effective retrieval system, and the results demonstrate clear improvements in performance compared to the state-of-the-art methods.
2. The framework is validated through extensive experiments on the M-BEIR benchmark, where U-MARVEL outperforms other methods in multiple retrieval tasks. It also shows strong zero-shot generalization, which is a critical aspect of real-world applications.

**Weaknesses:**

1. Retrieval and reranking are common pipelines in information retrieval. The authors apply this framework to MLLMs, seemingly leveraging the powerful capabilities of MLLMs to obtain better embeddings. The authors should further clarify the differences and novelty of the MLLM-based framework compared to traditional retrieval frameworks.
2. The authors use Qwen2-VL-7B as the base model for experiments. However, the performance of this retrieval framework on other MLLMs is unclear. The authors should further validate the generalization ability of this retrieval framework.
3. From Table 3 in the paper, I do not see a significant gain from the progressive transition design. The authors should further analyze this in the rebuttal.

**Questions:**

Please refer to the weaknesses.

---

> ### Author Response · Authors · 2025-11-26
>
> > Response to W1:
>
> We thank the reviewer for this insightful question. While the high-level "Recall-then-Rerank" pipeline is indeed a classic paradigm in information retrieval, applying it to MLLMs presents unique challenges that distinguish our framework from traditional approaches (e.g., BERT-based DPR or standard CLIP-based methods). The core novelty of U-MARVEL lies in bridging the paradigm gap between generative MLLMs and discriminative retrieval tasks. Specifically, we clarify the differences and our methodological novelties as follows:
>
> 1. **Architecture Adaptation (Generative vs. Discriminative):**
>
>    - **Traditional:** Traditional retrievers typically use Encoder-only architectures (e.g., BERT, ResNet) which naturally process inputs bi-directionally to extract holistic representations.
>
>    - **Our Novelty:** MLLMs are Decoder-only, autoregressive models designed for next-token prediction. A key challenge is adapting this causal architecture for effective embedding learning. Our work systematically proves that converting causal attention to Bidirectional Attention combined with Mean Pooling (Finding 1) and utilizing a Masked Instruction mechanism (Finding 2) significantly outperforms the standard "Last Token" approach often used in LLM adaptations. This is a specific architectural innovation tailored for Decoder-only MLLMs.
>
> 2. **Training Paradigm (Direct Fine-tuning vs. Progressive Transition):**
>
>    - **Traditional:** Traditional embedding models are often fine-tuned directly on target retrieval datasets.
>
>    - **Our Novelty:** We identify that directly fine-tuning MLLMs on complex multimodal instructions leads to suboptimal alignment. To address this, we introduce a Progressive Transition strategy, which guides the model through "Text Retrieval --> Cross-modal Alignment --> Instruction-tuned Retrieval." This step-wise recipe is uniquely designed to transition a general-purpose generative model into a specialist retriever without losing its pre-trained capabilities.
>
> 3. **Efficiency & Distillation (Cascaded vs. Unified):**
>
>    - **Traditional:** Classical pipelines often rely heavily on the Reranker for final performance, accepting high latency.
>
>    - **Our Novelty:** While we utilize a Reranker during training, our novelty lies in the Improved Distillation method (Finding 6). Unlike standard distillation, we propose a resource-efficient method that confines distillation to positive and hard negative samples. This allows us to compress the reasoning power of the MLLM Reranker back into the Retriever, enabling U-MARVEL to achieve state-of-the-art performance using single-stage inference, thus solving the efficiency bottleneck inherent in MLLM-based retrieval.

---

> > ### Author Response · Authors · 2025-11-26
> >
> > > Response to W2:
> >
> > We thank the reviewer for this insightful suggestion. We agree that validating the scalability and generalization of our findings across different model sizes is crucial.
> >
> > 1. **Prevalence of 7B Models in Current Literature:** We primarily focused on the 7B parameter scale because it currently serves as the standard benchmark for state-of-the-art MLLM-based retrieval methods (e.g., LamRA, MM-Embed, GME). Aligning with this scale allowed for fair and direct comparisons with existing literature.
> >
> > 2. **Generalization Verification on Qwen3-VL-4B (New Experiment):** To address your concern regarding whether our findings hold for different model sizes, we have extended our framework to the Qwen3-VL-4B model. As detailed in Table 16 and the newly added "Analysis on Generalization Ability of U-MARVEL" in Appendix D, our observations are as follows:
> >
> >    - **Consistent Findings:** The experimental results on the 4B model align perfectly with our conclusions from the 7B model. We observed steady performance gains across all three proposed stages (Progressive Transition, Hard Negative Mining, and Distillation), confirming that our training recipe is robust and not specific to a single architecture or size.
> >
> >    - **Superior Performance with Fewer Parameters:** Remarkably, despite having significantly fewer parameters, the Qwen3-VL-4B model trained with U-MARVEL achieves a Local Average Recall of 58.8% and a Global Average Recall of 56.2% on M-BEIR. This establishes a new state-of-the-art, significantly outperforming the previous best method, LamRA-Ret (Local: 56.6%, Global: 54.9%), which utilizes a larger 7B backbone.
> >
> >    - **Implication on Instruction Following:** Smaller models typically struggle more with instruction following compared to larger ones. The fact that U-MARVEL enables a 4B model to outperform a 7B SOTA baseline (LamRA-Ret) strongly suggests that our method effectively enhances the instruction-following and retrieval capabilities of the backbone, regardless of its size.
> >
> > 3. **Future Exploration on Larger Models:** We acknowledge the value of testing on significantly larger models (e.g., 32B or 70B) to further verify the upper limits of our approach. Due to the time constraints of the rebuttal period, we were unable to complete training on these larger scales. However, given the positive scaling behavior observed from 4B to 7B, we are optimistic about the performance on larger models and plan to include these experiments in future work.

---

> > > ### Author Response · Authors · 2025-11-26
> > >
> > > > Response to  W3:
> > >
> > > We thank the reviewer for carefully examining our ablation studies. We acknowledge that the incremental gain in the final step (ID-1 to ID-2) appears modest. However, we argue that the cumulative impact of the Progressive Transition strategy is substantial and methodologically critical when viewed holistically. We offer the following analysis to clarify the significance:
> > >
> > > 1. **Substantial Cumulative Improvement:** The "Progressive Transition" design encompasses the entire pipeline from the baseline (ID-0) to the final stage (ID-2).
> > >
> > >    - As shown in Table 3, comparing the complete strategy (ID-2) against the standard Instruction Tuning baseline (ID-0), we achieve an absolute improvement of 1.9% in the Global setting (53.9% --> 55.8%) and 1.1% in the Local setting (56.6% --> 57.7%).
> > >
> > >    - In the context of the M-BEIR benchmark, a nearly 2% improvement on the challenging Global metric is a significant margin, verifying that the stepwise adaptation is far superior to direct fine-tuning.
> > >
> > > 2. **Critical Role of Text Adaptation (ID-0 --> ID-1):** The most significant jump occurs during the first phase of our progressive design (incorporating text-only retrieval), which yields a 1.6% gain in Global Avg. As analyzed in Section 3.1.3, transitioning from the LLM's native causal attention to the bidirectional attention required for retrieval initially degrades the semantic quality of the text embeddings. The text-only fine-tuning (ID-1) strengthens the text encoder's semantic representation under the new attention mechanism. A robust text encoder serves as a necessary "semantic anchor"; without this foundation, the model struggles to align the visual encoder effectively because the textual targets themselves are suboptimal when directly exposed to complex multimodal instructions.
> > >
> > > 3. **Further Alignment via Cross-modal Data (ID-1 --> ID-2):** While the subsequent gain from ID-1 to ID-2 (+0.3% Global) is smaller, it represents a crucial fine-grained alignment of the visual encoder with the text encoder, as noted in our discussion. This step ensures the model is robust across diverse modalities, bridging the gap from single-modality to multi-modality tasks.
> > >
> > > In summary, the Progressive Transition design is not merely about the final incremental step but is a holistic recipe that delivers a 1.9% total performance gain, ensuring the effective adaptation of decoder-only MLLMs to embedding tasks.

---

### Official Review · Reviewer_kXJc · 2025-11-07

**Soundness:** 4
**Presentation:** 3
**Contribution:** 2
**Rating:** 4
**Confidence:** 4

**Summary:**

The authors perform a systemic investigation to identify an optimized configuration for universal multimodal retrieval (UMR) with a 'general' MLLM-based pipeline. Specifically, they finetune a Qwen2-VL-7B-Instruct model with LoRA as the retriever-reranker embedding model (with distillation) using contrastive learning (InfoNCE) on the M-BEIR dataset (Figure 1b). Based on this 'basic' architecture, they explore three primary questions: (1) how to adapt decoder-only MLLMs into instruction-aware embedding function, (2) how to train embedding models with contrastive learning, and (3) the value of distillation in recall-then-rerank methods.

For (1), the first question addressed is whether to use {last token, mean token}, {unidirectional, bidirectional}, and the use of a compression prompt instruction (i.e., one-word) in extracting embeddings from the MLLM [bidirectional, mean pooling wins]. The second question addressed for (1) is instruction inclusion and their masking out during mean pooling [masking out instructions during mean pooling wins]. The third question is testing progressive transition in fine-tuning MLLMs with step wise training (instruction tuning -> text-only retrieval (NLI) -> text-image retrieval (CC3M)) [it works].

For (2), the first question asked is the hyperparameter grid search over {batch size, learnable temperature, learning rate} [increasing batch size with appropriate rate scaling and learnable temperature parameters enhance effectiveness]. The second question asked for (2) explores hard negative mining, in-batch negatives, and negative filtering when training InfoNCE [in-batch negatives and filtered top-k hard negatives wins (there are some details on the hard negative filtering strategy)].

For (3), they show that reranker distillation reduces computational requirements and increases feature diversity (i.e., effective distillation via a teacher-student model). This section has a lot of technical details (especially if also considering the appendices) [the method distills the teacher model with minimal performance degredatation]

Put all of this together, and you have U-MARVEL, which is shown to outperform several single model (e.g., LamRA-retriever) and recall-then-rerank (e.g., LamRA) models on M-BEIR in local and global pool settings. On zero-shot settings with unseen datasets, U-MARVEL outperforms all single model settings, is competitive with LamRA on the recall-then-rerank setting for image/text retrieval, and outperforms all baseline models on text-to-video retrieval benchmarks.

**Strengths:**

Strengths of this work include:
- Variants of the base architecture captures most state-of-the-art systems for universal multi-modal retrieval with VLLM backbones.
- The configuration/hyperparameter search space addresses several relevant questions and provides good evidence for particular design choices.
- The resulting empirical performance is strong across multiple datasets and settings and is compared to multiple recent strong baseline systems.

**Weaknesses:**

Weaknesses of this work include:
- While this is partially attributable to space limitations, it was difficult to assess the soundness of the precise method without reading the appendices (I would make some different choices in terms of details, etc.). In the same vein, many of the empirical results in the appendices are useful and not incorporated into the discussion. Finally, with respect to the empirical results, the discussion/analysis is mostly 'just' restating the tables with limited interpretation.
- Maybe I am missing a description of the notation in the empirical results, but I wasn't able to discern statistical significance. Many of the results don't appear statistically significant.
- As the authors are aware, limiting experiments to 7B models is limiting as I would expect that the findings may differ for larger models (specifically with respect to instruction following). While these are comparable to other models in the literature, as this work follows these, it is going to be necessary to make these comparisons soon.
- Modulo the ranker distillation details, there is limited methodological novelty. This is a classic "optimize all the pieces" of a architecture to create a definitive strong baseline for this moment in time that more innovative variants can directly compare to.

**Questions:**

Some questions of mine would include:
- Statistical significance of the empirical results
- Based on your experience in conducting all of these experiments, what are some directions you are thinking about for further improvements that may lead to notable improvements (if any)?
- (For an appendix), were there any additional directions explored that led to inconclusive or interesting negative results?

---

> ### Author Response · Authors · 2025-11-25
>
> > Response to W1:
>
> We sincerely appreciate the reviewer’s constructive feedback regarding the organization of our methodology and the depth of our discussion. We agree that relegating key details to the appendix hindered the assessment of our method's soundness and that the initial discussion lacked sufficient interpretive depth.
>
> We have extensively revised the manuscript to address these points. All significant revisions have been marked in red in the updated PDF. Specifically, we have relocated core mathematical formulations to the main text, provided granular implementation details, and significantly expanded our analysis to focus on underlying mechanisms rather than merely restating numerical results. Detailed responses to specific concerns are provided below:
>
> **Comment 1:**   “While this is partially attributable to space limitations, it was difficult to assess the soundness of the precise method without reading the appendices... I would make some different choices in terms of details...”
>
> **Response:**   Thank you for pointing this out. We understand that the soundness of the method cannot be properly evaluated if critical details are buried in the appendix. We have moved essential definitions and algorithmic details from Appendix C back to Section 3 (Methodology) to ensure the paper is self-contained:
>
> -  **Mathematical Formulations:**  The formal definitions of the InfoNCE loss with learnable temperature (Eq. 1) and the Distillation loss (Eq. 2) are now presented directly in the main text.
>
> - **Specific Implementation Details:** We have expanded the text to explicitly describe key hyperparameters that were previously omitted. This includes the specific threshold settings used for filtering false negatives in Section 3.2.2 and the configuration of the weighting parameter α for the recall-rerank fusion in Section 3.3.
>
> **Comment 2:**   “In the same vein, many of the empirical results in the appendices are useful and not incorporated into the discussion.”
>
> **Response:**   We agree that the results in the appendices offer valuable insights that were previously overlooked. We have integrated these findings into Section 4 and Appendix D, and added new experiments to strengthen our claims:
>
> - **Integration of Appendix Results:**  We now explicitly discuss the Ablation Study (Table 15) in Section 4.1 to demonstrate the specific contribution of each training stage. We also incorporated the Global Pool settings (Table 17) into the main discussion to highlight U-MARVEL’s robustness in large-scale retrieval scenarios.
>
> - **New Generalization Experiments:** To address the generalizability of our approach, we have added new experiments using different backbones to verify that our findings hold across varying model architectures.
>
> - **Reproducibility:**  We have included a dedicated description regarding our protocols for controlling experimental randomness, ensuring the reliability of the reported results.
>
> **Comment 3:**  “Finally, with respect to the empirical results, the discussion/analysis is mostly 'just' restating the tables with limited interpretation.”
>
> **Response:**  This is a valid critique. We have rewritten the analysis in Sections 4.1 and 4.2 to move beyond listing performance gains and instead interpret the reasons behind them:
>
> - **Mechanism Analysis:**  We have added deep-dive analyses on specific components, including an interpretation of why the learnable temperature coefficient is critical for large-batch training (Finding 4) and an analysis of how different data sources (e.g., NLI vs. CC3M) contribute to the model's capability evolution during the Progressive Transition stage (Finding 3).
>
> - **Efficiency vs. Effectiveness:** Instead of merely stating that U-MARVEL outperforms LamRA, we now analyze the trade-off between efficiency and effectiveness, explaining how our improved distillation strategy allows a single-stage model to match the performance of computationally expensive cascade systems.

---

> > ### Author Response · Authors · 2025-11-26
> >
> > > Response to W2:
> >
> > We thank the reviewer for raising this important point regarding the rigor of our empirical evaluation.
> >
> > 1. **Regarding Experimental Setup and Reproducibility**: We would like to clarify that, following standard practices in large-scale MLLM fine-tuning (where computational costs for multiple runs are often prohibitive), we utilized fixed random seeds for all experiments. This ensures that our reported results are deterministic and fully reproducible. We have added an explicit statement regarding this setup in Appendix C (Implementation Details) to avoid ambiguity.
> >
> > 2. **Regarding Statistical Significance**: While we did not perform multiple runs to compute standard deviations due to the aforementioned computational constraints, we argue that the performance gains achieved by U-MARVEL are statistically significant and practically meaningful for the following reasons:
> >
> >    - **Large Performance Margin**: As demonstrated in Table 7, U-MARVEL achieves an average improvement of 6.6% (Local) and 5.8% (Global) over the previous state-of-the-art (LamRA-Ret). This margin is substantial and far exceeds the typical variance observed in fine-tuning experiments (which is usually within 0.5-1.0%).
> >
> >    - **Consistency Across Tasks**: Statistical significance can also be inferred from the consistency of the results. U-MARVEL outperforms the strongest baseline in 15 out of the 16 retrieval tasks evaluated on the M-BEIR benchmark. The probability of achieving such consistent dominance by random chance is negligible.
> >
> > ---
> >
> > > Response to W3:
> >
> > We acknowledge the reviewer’s valid point regarding the limitation of experimenting solely with 7B models. We would like to address this concern from three perspectives:
> >
> > 1. **Fair Comparison with State-of-the-Arts**: As noted by the reviewer, our choice of the 7B parameter scale aligns with the current mainstream literature in Universal Multimodal Retrieval. Key baselines such as LamRA, MM-Embed, and VLM2Vec are all built upon 7B backbones. Using the same model size ensures that the performance gains reported in Table 7 are strictly attributable to our proposed framework (U-MARVEL) rather than advantages in model capacity.
> >
> > 2. **Generalizability of Findings**: Regarding the concern about instruction following, we believe our core findings—specifically the "Progressive Transition" strategy and "Masked Instruction" mechanism—are methodological principles rather than scale-dependent heuristics. While larger models indeed possess stronger intrinsic instruction-following capabilities, the challenge of adapting a generative objective to a discriminative embedding space remains universal. Our proposed training recipes are designed to effectively bridge this gap, a principle we expect to hold for larger models as well.
> >
> > 3. **Generalization Verification on Qwen3-VL-4B (New Experiment)**: To address your concern regarding whether our findings hold for different model sizes, we have extended our framework to the Qwen3-VL-4B model. As detailed in Table 16 and the newly added "Analysis on Generalization Ability of U-MARVEL" in Appendix D, our observations are as follows:
> >
> >    - **Consistent Findings**: The experimental results on the 4B model align perfectly with our conclusions from the 7B model. We observed steady performance gains across all three proposed stages (Progressive Transition, Hard Negative Mining, and Distillation), confirming that our training recipe is robust and not specific to a single architecture or size.
> >
> >    - **Superior Performance with Fewer Parameters**: Remarkably, despite having significantly fewer parameters, the Qwen3-VL-4B model trained with U-MARVEL achieves a Local Average Recall of 58.8% and a Global Average Recall of 56.2% on M-BEIR. This establishes a new state-of-the-art, significantly outperforming the previous best method, LamRA (Local: 56.6%, Global: 54.9%), which utilizes a larger 7B backbone.
> >
> >    - **Implication on Instruction Following**: Smaller models typically struggle more with instruction following compared to larger ones. The fact that U-MARVEL enables a 4B model to outperform a 7B SOTA baseline(LamRA-Ret) strongly suggests that our method effectively enhances the instruction-following and retrieval capabilities of the backbone, regardless of its size.
> >
> > 4. **Future Exploration on Larger Models**: We acknowledge the value of testing on significantly larger models (e.g., 32B or 70B) to further verify the upper limits of our approach. Due to the time constraints of the rebuttal period, we were unable to complete training on these larger scales. However, given the positive scaling behavior observed from 4B to 7B, we are optimistic about the performance on larger models and plan to include these experiments in future work.

---

> > > ### Author Response · Authors · 2025-11-26
> > >
> > > > Response to W4:
> > >
> > > We thank the reviewer for recognizing our work as a "definitive strong baseline" for the field. We agree that establishing a robust, rigorous baseline is crucial for the progress of universal multimodal retrieval, specifically to prevent future research from comparing against suboptimal implementations.
> > >
> > > However, we respectfully suggest that the "methodological novelty" of this work extends beyond the distillation mechanism. We view our contribution not merely as "optimizing pieces," but as systematically uncovering the critical design principles that have been largely fragmented or misunderstood in prior literature. Specifically:
> > >
> > > 1. **Novelty in Discovery**: Scientific novelty lies not only in proposing new architectures but also in discovering why certain designs work. For instance:
> > >
> > >    - **Bidirectional vs. Last Token**: We provide empirical evidence challenging the "Last Token" paradigm used in recent SoTAs, proving that bidirectional attention with mean pooling is superior.
> > >
> > >    - **Masked Instruction**: We introduce a "Masked Instruction" mechanism during pooling (Finding 2), which is a specific architectural modification to mitigate the bias introduced by instruction tokens.
> > >
> > > 2. **Novelty in Training Recipes**: Our Progressive Transition strategy and the Threshold-based False Negative Filtering are not standard "out-of-the-box" optimizations. They are specific methodological solutions designed to address the unique challenges of adapting generative MLLMs to discriminative embedding tasks (e.g., preventing model collapse during hard negative mining).
> > >
> > > 3. **Novelty in Distillation (As noted)**: As the reviewer kindly acknowledged, our improved distillation method is a significant innovation. It reduces the computational complexity of the teacher-student alignment from $\mathcal{O}(Nn)$ to $\mathcal{O}(Nk)$, making the distillation of sophisticated rerankers into single-stage retrievers computationally feasible for the first time in this context.
> > >
> > > In summary, while we assemble existing building blocks (like InfoNCE or LoRA), the specific arrangement, modification, and rigorous evaluation of these blocks constitute a novel methodological framework (U-MARVEL) that significantly advances the state-of-the-art.

---

> > > > ### Author Response · Authors · 2025-11-26
> > > >
> > > > >Response to  Q1:
> > > >
> > > > **Please refer to W2.**
> > > >
> > > > ---
> > > >
> > > > > Response to Q2:
> > > >
> > > > Based on our experimental findings, we have identified two promising directions to further enhance performance:
> > > >
> > > > 1. **Dynamic, Query-Aware Visual Compression**:  A limitation of current architectures is "static compression": the ViT compresses visual information into tokens independently of the text query, potentially discarding details crucial for specific instructions. To address this, we plan to redesign the feature compression module to be text-driven. Specifically, we aim to introduce text-conditional dynamic convolution kernels into the ViT’s downsampling layers. This mechanism would allow the visual encoder to actively preserve fine-grained visual details aligned with the user's textual intent, rather than generating a generic representation.
> > > >
> > > > 2. **Sophisticated Feature Aggregation**:  While our current mean pooling strategy outperforms the last-token mechanism, it treats all tokens equally, which may dilute key local features. We plan to investigate more advanced aggregation strategies to capture token importance and inter-correlations. Directions include Self-Attention Pooling to adaptively weigh core features and suppress noise, and Cross-Attention Pooling to explicitly model fine-grained interactions between textual and visual tokens, thereby generating more discriminative multimodal embeddings.
> > > >
> > > > ---
> > > >
> > > > > Response to Q3:
> > > >
> > > > We thank the reviewer for this insightful suggestion. We agree that discussing negative results contributes significantly to the community. We have indeed explored several directions that yielded inconclusive or negative outcomes, as follows:
> > > >
> > > > 1. **Limited Zero-Shot Generalization in Video Reranking**: While our embedding model demonstrates strong generalization, we observed that our reranker model exhibits suboptimal performance on zero-shot video retrieval tasks. We attribute this to the reranker's architecture being trained primarily on static image-text pairs, lacking the temporal modeling capabilities required to capture the dynamic information in videos effectively, shown in Sec. 4.2.
> > > >
> > > > 2. **Counter-Intuitive Impact of Dense Captions (ShareGPT4V)**: During the progressive training stage, we attempted to incorporate high-quality image-text data from ShareGPT4V, which features highly detailed captions. Surprisingly, whether used alone or mixed with CC3M, this data led to a performance degradation on the M-BEIR benchmark. We hypothesize that the significant distribution shift between the dense, descriptive captions of ShareGPT4V and the concise, query-oriented nature of retrieval tasks (M-BEIR) hindered the model's alignment for the target retrieval objective. We have added these findings and analyses in Appendix D to provide a more comprehensive view of our exploration.

---

### Author Response · Authors · 2025-11-29
**General Response: Summary of Revisions and Additional Experiments - 1 / 2**

We sincerely thank all reviewers for their insightful and constructive comments. We have carefully considered all feedback and have significantly revised the manuscript to improve clarity, technical depth, and comprehensiveness. The major changes in the revised paper (highlighted in **red**) are summarized below:

**1. Clarification of Technical Details and Hyperparameters (Sections 3 & Appendix C)**

To address the concern that "it is difficult to assess the reasonability of the method without reading the appendix," we have moved critical mathematical formulations and hyperparameter details from the appendix or implicit assumptions directly into the main text:

* **Formal Definitions:** We have explicitly defined the loss functions in the main text, including the **InfoNCE loss** (Eq. 1) in Section 3 and the **Distillation loss** (Eq. 2) in Section 3.3.

* **Key Hyperparameters:** We have provided precise values for key parameters to ensure the method is self-contained. For instance:

    * In **Section 3.2.2**, we specify the configuration for hard negative mining, including the filtering threshold ($0.7$) and the number of hard negatives ($k=5$).

    * In **Section 3.3 and Appendix C**, we explicitly state the linear interpolation formula for the recall-rerank fusion: $S_{multi} = \alpha \cdot S_{recall} + (1 - \alpha) \cdot S_{rerank}$, and specify that $\alpha$ is fixed at $0.5$.

**2. Deepened Analysis and Interpretation (Sections 3 & 4)**

We agree with the feedback that previous discussions were occasionally limited to restating empirical results. We have rewritten the analysis sections to provide **in-depth interpretations** of *why* certain phenomena occur, rather than just reporting *that* they occur:

* **Section 3.1.2:** We explain that masking instruction tokens during mean pooling improves performance by mitigating calculation bias, as the query already incorporates instruction information via self-attention.

* **Section 3.2.1:** We analyze the interaction between batch size and learning rate, explaining the necessity of the scaling rule. We also discuss why a **learnable temperature** creates a sharper probability distribution that adapts better to training dynamics compared to a fixed temperature.

* **Section 3.2.2:** We provide a rationale for the "Hard Negative Mining" strategy, explaining that naively maximizing loss on all hard negatives leads to model collapse due to false negatives in the dataset.

* **Section 4.1:** We analyze the efficiency-performance trade-off, highlighting how our distillation strategy successfully transfers the reranker's discriminative power into the embedding space.

---

> ### Author Response · Authors · 2025-11-29
> **General Response: Summary of Revisions and Additional Experiments - 2 / 2**
>
> **3. New Supplementary Experiments and Analysis (Appendices D & E)**
>
> To further strengthen the empirical validation, we have added comprehensive new analyses:
>
> * **Analysis on Training Data for Progressive Transition (Appendix D, Table 13):** We added experiments comparing different data sources (e.g., ShareGPT4V, TART). We analyze why high-quality caption data (ShareGPT4V) failed to improve performance, attributing it to the distribution shift between exhaustive descriptions and retrieval-oriented queries.
>
> * **Analysis on Learnable Temperature of InfoNCE (Appendix D):** We demonstrate that a learnable $\tau$ consistently outperforms fixed settings (including manually tuned values) by dynamically adapting to the optimization landscape. It exhibits stable convergence ($\tau \approx 0.02$) across varying batch sizes and evolves stage-wise—notably dropping during Hard-negative Mining to enhance sensitivity—thereby ensuring superior performance and robust training stability.
>
> * **Analysis on Generalization Ability  of U-MARVEL (Appendix D, Table 16):** We extended our framework to a different backbone (**Qwen3-VL-4B**) to test generalization. The results confirm that our training recipe is robust and model-agnostic.
>
> * **Distillation Computational Cost Analysis (Appendix E, Table 14):** We added a rigorous mathematical analysis comparing the computational complexity of Traditional vs. Improved Distillation. This theoretical proof supports our claim that the improved method significantly reduces overhead (specifically in the Rerank phase) while increasing feature diversity.
>
> **4. Enhanced Reproducibility (Appendix C)**
>
> We have added a dedicated **Reproducibility** section in Appendix C, detailing our random seed settings (fixed to 42) and computing environment to ensure fair and deterministic comparisons.
>
> We believe these revisions directly address the reviewers' concerns regarding the depth of analysis and the clarity of method details. We hope these improvements make the paper stronger and more accessible.

---

### Meta-Review · Area_Chair_aH7f · 2026-01-04

**Summary:**

Summary of paper:
This paper introduces U-MARVEL, a unified framework for learning universal multimodal embeddings, tailored to universal multi-modal retrieval (UMR) tasks.
- Section 3 proposes the recipe for the U-MARVEL framework. Section 3.1 discusses adapting MLLMs that generate outputs in an autoregressive manner to embedding models that output a single embedding. Section 3.1.1 provides a comparison between two common approaches. Section 3.1.2 finds instruction masking during mean pooling to help. Section 3.1.3 proposes progressive transition for adapting decoder-only MLLMs to embedding models through an additional pre-training step.
- Section 3.2 analyzes the impact of training modifications. Section 3.2.1 studies the effect of batch size, learning rate scaling and temperature. Section 3.2.2 studies training with hard negative mining.
- Section 3.3 studies training a reranker model then distills the framework of recall-then-rerank to a student model. The distillation is done only on positive and hard negative samples.
- Section 4 provides comparisons with other state-of-the-art approaches.

Summary of strengths by reviewers:
- Reviewer SuPC said the problem is impactful and the proposed framework improves over M-BEIR particularly in single model retrieval.
- Reviewer WToA said the paper provides actionable and well-motivated findings and the paper provides details that would help in reproducibility.
- Reviewer 4154 said the paper successfully integrates multiple advanced techniques and the results demonstrate clear improvements and the framework is validated through extensive experiments.
- Reviewer kXJc said the unified framework captures most state-of-the-art systems,  the configuration/hyperparameter search provides good evidence for choices, and the empirical performance is strong.

**Reviewer Concerns:**

Reviewer SuPC:
- Writing clarity: Reviewer asked for some paragraphs to be revised to improve readability. The authors addressed it in their revision.
- Evidence strength: Reviewer raised concern about the significance of gains in some of the tables and suggested adding significance tests. Authors discussed the significance specifically in instruction masking and learning rate scaling.
- Motivations: The reviewer asked for justification of some of the choices in the method. Authors revised the paper to add citations for curriculum learning and pointed to existing results in Table 3 that would respond to reviewer’s concern.

Reviewer WToA
- Limited novelty: The reviewer found the work to be a combination of existing tricks in retriever training and ones ported from LLM to MLLM. The authors argued their contribution lies in the systematic discovery of why and how these components must be adapted to bridge the gap between generative MLLMs and discriminative retrieval tasks. The reviewer responded they still hold their opinion about the novelty.
- Insufficient detail on hard-negative filtering criteria: The reviewer asked for more details and authors provided.
- Questions and concerns regarding experimental design: The reviewer asked various questions about the details of experiments. The authors provided detailed responses to each question.

Reviewer 4154
- Novelty: The reviewer asked for novelty compared with traditional retrieval frameworks. The authors provided a detailed list of comparisons summarized in 1) Architecture Adaptation (Generative vs. Discriminative) 2) Training Paradigm (Direct Fine-tuning vs. Progressive Transition) 3) Efficiency & Distillation (Cascaded vs. Unified).
- Extension to other base model architectures: The reviewer raised concern about evaluating only on Qwen2-VL-7B. The authors argue that 7B models are currently standard in literature and for comparison to other methods while they also extend their results to Qwen3-VL-4B.
- Significance of gains in Table 3 (progressive transition): The authors argue that the overall total performance gain of 1.9% is significant that is the result of a holistic recipe.

Reviewer kXJc
- Results in the appendix: The reviewer suggested that some of the results in the appendix are useful and could be brought to the main body. The authors made changes to the paper and added more results from the appendix to the main body.
- Statistical significance: The reviewer raised concern that many of the results do not appear statistically significant. The authors point to the final comparison with prior SOTA and argue that the proposed framework achieves significant improvements and that it performs best in the majority of retrieval tasks.
- Limiting experiments to 7B models is limiting: The authors argued this is for fair comparison to existing methods but also provided results on Qwen3-VL-4B.
- Limited novelty: The reviewer argued that except for ranker distillation, the rest of the paper has limited novelty. The authors provided a detailed list of novelties in each part.

**Reviewer Scores:**

All reviewers gave the score of 4 (marginally below the acceptance threshold). Most concerns were addressed by adding details to the paper and improving the writing. The concern about novelty was discussed in detail in response to multiple reviewers. The AC acknowledges the reviewers’ concerns that the paper draws many ideas from existing literature. At the same time, bringing ideas together in a unified framework and achieving SOTA results is also a notable contribution. All reviewers have acknowledged the significance of the final results. As such, the AC recommends acceptance.

---

### Decision · Program_Chairs · 2026-01-26

Accept (Poster)